# Biofilm and Outer Membrane Vesicle Formation in ESKAPE Gram-Negative Bacteria: A Comprehensive Review

**DOI:** 10.3390/ijms26209857

**Published:** 2025-10-10

**Authors:** Giedrė Valdonė Sakalauskienė, Aurelija Radzevičienė

**Affiliations:** 1Department of Cosmetology, Faculty of Medicine, Kauno Kolegija Higher Education Institution, Pramonės pr. 20, LT-50468 Kaunas, Lithuania; 2Institute of Physiology and Pharmacology, Faculty of Medicine, Medical Academy, Lithuanian University of Health Sciences, 44307 Kaunas, Lithuania

**Keywords:** ESKAPE Gram-negative bacteria, outer membrane vesicles, biofilm, antimicrobial resistance, virulence

## Abstract

Antimicrobial resistance (AMR) is a growing global threat, exacerbated by the adaptive mechanisms of Gram-negative ESKAPE pathogens, which include biofilm formation and outer membrane vesicle (OMV) production. Biofilms create robust protective barriers that shield bacterial communities from immune responses and antibiotic treatments, while OMVs contribute to both defense and offense by carrying antibiotic-degrading enzymes and delivering virulence factors to host cells. These mechanisms not only enhance bacterial survival but also increase the virulence and persistence of infections, making them a significant concern in clinical settings. This review explores the molecular processes that drive biofilm and OMV formation, emphasizing their critical roles in the development of AMR. By understanding these mechanisms, new therapeutic strategies can be developed to disrupt these defenses, potentially improving the efficacy of existing antibiotics and slowing the spread of resistance. Additionally, the use of OMVs in vaccine development and drug delivery offers promising avenues for future research. Addressing these challenges requires a comprehensive approach, combining advanced research with innovative therapies to combat the escalating threat of AMR and improve patient outcomes.

## 1. Introduction

Antimicrobial resistance (AMR) represents a natural evolutionary process that has been greatly accelerated by the excessive and inappropriate use of antimicrobial agents. Over time, bacteria, viruses, fungi, and parasites adapt, leading to reduced efficacy or even complete failure of available treatments. As a result, infections become more difficult to manage, the likelihood of transmission increases, and the risk of severe disease and death rises [1,2]. In 2019, according to the global surveillance data, AMR was linked to almost 5 million deaths worldwide, including approximately 1.27 million cases directly caused by drug-resistant infections [3]. Forecasts indicate that by 2050, AMR-related mortality could reach 10 million annually, surpassing deaths from cancer [4]. The economic burden is also substantial: the World Bank projects that by 2030, the global GDP loss attributable to AMR may approach 3.4 trillion US dollars per year, and by 2050, AMR could reduce the world’s GDP by about 3.8%, with an additional 24 million individuals potentially falling into extreme poverty within the next decade [1,5,6].

The occurrence of AMR is not uniform across bacterial species [7,8]. The World Health Organization (WHO) has categorized a group of particularly problematic microorganisms, collectively referred to as “ESKAPE” pathogens, due to their pronounced multidrug resistance and virulence [9,10]. This group comprises *Enterococcus faecium* (*E. faecium*), *Staphylococcus aureus* (*S. aureus*), *Klebsiella pneumoniae* (*K. pneumoniae*), *Acinetobacter baumannii* (*A. baumannii*), *Pseudomonas aeruginosa* (*P. aeruginosa*), as well as *Enterobacter* species (*Enterobacter* spp.), which together account for the majority of hospital-acquired infections [9,11]. Based on the clinical and epidemiological studies it was revealed that the coronavirus disease (COVID-19) [12] pandemic has further complicated efforts to combat AMR, as the increased use of antimicrobials to treat secondary bacterial infections or as prophylaxis during the pandemic has exacerbated resistance issues worldwide [13,14,15].

A systematic review of clinical data covering 23 studies released between 2019 and 2022 reported particularly elevated resistance rates in *A. baumannii* and *K. pneumoniae* (Table 1) [13,15]. In addition, *Enterobacter* spp., particularly the *E*. *cloacae* complex, displayed some concerning antimicrobial resistance trends during the COVID-19 pandemic. Several outbreaks of carbapenemase-producing strains such as Verona Integron-Borne Metallo-β-Lactamase (VIM) [16] and New Delhi metallo-β-lactamase-1 (NDM-1) [17] were reported in European COVID-19 intensive care units (ICUs) [18,19]. According to national surveillance data from CHINET antimicrobial resistance surveillance (2015–2021), analysis of isolates from 53 hospitals showed that carbapenem resistance among *Enterobacter* spp. reached 10.0%, with data indicating a progressive rise throughout the seven-year monitoring period [20], possibly influenced by increased antibiotic use during the COVID-19 pandemic [21]. By contrast, clinical hospital surveillance data from King Fahad Medical City in Riyadh indicated a decline in carbapenem-resistant *E. cloacae*, with rates falling from 48.36% in 2019 to 38.0% in 2020 and 37.6% in 2021 (Table 1) [22]. In addition, among Gram-positive ESKAPE pathogens, *S. aureus* and *E. faecium*, exhibited notable resistance. The included studies originated from diverse regions worldwide, with several conducted in Asia, including India, Iran, Saudi Arabia, China, Turkey, and Pakistan [13,14]. Although the analysis of 1959 unique isolates did not yield statistically significant differences, observed trends suggested higher levels of AMR in non-European countries and ICU settings [23]. European contributions came from Italy, Switzerland, Greece, and Serbia, while additional data were reported from Egypt, Indonesia, and the United States [13,14]. While these regions previously reported low to moderate resistance rates, a marked rise in resistance to last-resort antibiotics was documented between 2018 and 2020, with particularly concerning trends in polymyxin resistance [13,15]. However, a precise assessment of antimicrobial resistance changes before and after the pandemic across different regions requires continuous surveillance and rigorously designed studies [24].

The convergence of rising AMR in ESKAPE pathogens and the COVID-19 pandemic has been described as a “syndemic” [25,26,27]. For instance, hospital environmental studies from Mexico reported that during the pandemic, Gram-negative bacteria, particularly *A. baumannii*, were frequently detected on the surfaces of medical equipment, facilitating their spread in healthcare settings [25,28,29].

Additionally, *P. aeruginosa* and *A. baumannii* were identified as major causes of ventilator-associated pneumonia (VAP) in COVID-19 patients in Mexico [30,31]. However, a systematic review of 18 clinical studies analyzing VAP incidence and treatment during the second and third waves of the pandemic found that *P. aeruginosa*, *Enterobacterales* spp., and *S. aureus* were the predominant pathogens associated with VAP in COVID-19 patients globally [32].

Thus, these pathogens are the primary causes of dangerous infections, particularly among children, immune-deficient, and critically ill individuals [7,33]. These bacteria possess sophisticated mechanisms for evading antimicrobial agents, including drug inactivation, target site modification, and reduced drug accumulation [9,33]. Among the mechanisms contributing to AMR in ESKAPE pathogens, biofilm formation and the production of outer membrane vesicles (OMVs) are particularly significant [9,34].

Biofilms consist of structured bacterial communities that attach to surfaces and become surrounded by a self-generated matrix, providing protection against host immunity and antimicrobial agents [33,35,36]. Additionally, OMVs are spherical bilayered nanoparticles produced by Gram-negative bacteria that carry antibiotic-degrading enzymes and virulence factors, contributing to both resistance and pathogenicity [34]. Due to their biological characteristics and their roles in bacterium-bacterium and bacterium-host interactions, OMVs can also serve as vesicle-based vaccines and drug-delivery platforms [37]. Given the critical roles these two mechanisms play in AMR, this review emphasizes the importance of biofilm formation and OMV production in the context of Gram-negative ESKAPE pathogens.

## 2. Biofilms in Gram-Negative ESKAPE Bacteria

### 2.1. The Main Aspects of Biofilm Formation

Bacterial biofilms are structured microbial communities composed of single or multiple bacterial strains. They are frequently found on hospital equipment, body tissues, industrial surfaces, and food processing facilities, as well as in natural environments. Nearly all bacterial species can form biofilms, which complicate infection treatment and promotes persistence. Due to their increased resistance to standard antibiotics, biofilms are major drivers of infections associated with medical devices and tissue surfaces, representing a substantial challenge for healthcare systems worldwide [38,39].

A biofilm consists of approximately 10% biomass and 90% water. Polysaccharides, known as exopolysaccharides, make up 50–90% of its organic content and serve as key components of the extracellular polymeric substance (EPS) matrix [38,40]. Biofilm formation progresses through distinct phases: initial bacterial attachment (which may be reversible or irreversible), aggregation and microcolony formation, maturation, and, ultimately, dispersion or detachment (Figure 1) [38,39,40].

Bacterial communication, referred to as quorum sensing (QS), is essential for biofilm development. This process depends on the synthesis and recognition of extracellular signaling molecules called autoinducers. QS influences multiple stages of biofilm progression, from initial formation to dispersal, and is regulated in response to microbial population density [41]. When signaling molecule concentrations exceed a defined threshold, they bind to intracellular receptors, initiating the expression of target genes. This activation governs various physiological processes, including adaptation to environmental conditions, motility, adhesion, virulence factor production, and sporulation [41,42,43,44,45].

Typically, QS systems include a histidine sensor kinase located in the membrane and a cytoplasmic response regulator that functions as a transcriptional regulator controlling gene expression [41,46]. Beyond their role in bacterial regulation, quorum-sensing molecules also modulate host immune responses [41,47,48], contributing to the development of antibiotic resistance [41,49,50].

Phase I. Planktonic bacteria reversibly adhere to the biotic or abiotic surface. Attachment of planktonic cells depends on bacterial appendages, surface properties, and environmental factors, with adhesion mediated by van der Waals, electrostatic, hydrophobic, and protein interaction [38]. OMVs modulate physicochemical surface properties and enhance early biofilm establishment by influencing bacterial adhesion and surface hydrophobicity [51].

Phase II. Bacterial cells irreversibly attach on the surface, when planktonic bacteria enter a sessile state. Once bacteria adhere to a specific surface, adhesion becomes irreversible through the production of EPS and the accumulation of multilayered cell clusters. This transition is accompanied by QS signaling, which coordinates intercellular communication via chemical messengers and facilitates biofilm development. During this stage, cells lose motility and undergo physiological and structural adaptation [38]. OMVs associate with eDNA and adhesin proteins, providing structural support and strengthening the nascent matrix [52,53].

Phase III. Bacterial cells aggregate, multiply and divide, forming the microcolonies. Extensive secretion of the EPS matrix—regulated by signals such as c-di-GMP—drives bacterial multiplication and microcolony formation. The matrix acts as a structural binder, enabling stable adhesion, intercellular communication, nutrient exchange, and protection [38]. OMVs bind with eDNA and extracellular proteins, stabilizing the biofilm matrix and enhancing biomass accumulation [54,55].

Phase IV. Bacteria form a mature biofilm. Attached cells release signaling molecules that activate biofilm-associated genes and enhance virulence. The secretion of EPS stabilizes the biofilm and protects embedded cells against antimicrobials. As multiple cell layers accumulate, microcolonies expand into three-dimensional structures. Maturation occurs in two stages: first, cell-to-cell contact and autoinducer production; second, further thickening of microcolonies up to ~100 µm. During this process, QS coordinates gene and protein expression at the community level, fostering cooperative interactions and strengthening biofilm resilience [38]. OMVs further stabilize the matrix and reinforce biomass, while also transporting virulence and resistance factors that support biofilm endurance [56].

Phase V. The mature biofilm matrix disintegrates, and bacteria are released to colonize new surfaces. Bacteria detach from the mature biofilm and spread to colonize new surfaces. Detachment can be active, driven by bacterial responses to stress (e.g., nutrient limitation, antimicrobials) and mediated by matrix-degrading enzymes, or passive, caused by external forces such as fluid shear. Reduced levels of c-di-GMP favor separation, while environmental conditions (temperature, pH, oxygen, nutrient availability) further modulate dispersal [38]. During dispersal, OMVs are enriched with proteases, lipases, and DNases, which contribute to matrix breakdown and the release of free-floating planktonic bacteria [55,57].

Bacteria within biofilms exist in a sessile state and orchestrate key physiological activities within their environment. These communities exhibit unique growth dynamics, gene expression patterns, and transcriptional and translational rates. Their functional traits evolve in response to microenvironmental factors such as elevated osmolarity, nutrient limitation, and high cell density. Consequently, biofilm structures develop pronounced viscoelastic properties, resembling rubber-like behavior [38,40]. Therefore, biofilm thickness can vary depending on environmental conditions [38,58].

Bacteria capable of forming biofilms gain protection from other pathogens, improve their survival, and acquire the ability to spread and colonize new surfaces [38,58]. The EPS matrix in biofilms acts as a barrier, hindering antibiotics from achieving effective bactericidal concentrations. Bacteria within biofilms demonstrate significantly greater resistance to antibiotics compared to their planktonic counterparts, partly due to the biofilm’s protective structure and physiological factors like slower growth rates [38,59].

Reduced antimicrobial susceptibility in biofilm-forming bacteria arises through diverse processes, such as limited antibiotic permeability within the extracellular matrix, biochemical interactions that impair drug activity, enzymatic inactivation, metabolic shifts, genetic changes, and enhanced efflux pump activity removing antibiotics from the cells. This type of resistance differs from innate resistance, involving unique molecular adaptations, such as the formation of persistent cells that withstand antibiotic treatment and genetic changes that promote survival in hostile environments [38,59].

High mutation rates within biofilm communities accelerate the development of resistance mechanisms, including the production of enzymes that neutralize antibiotics. Biofilms also facilitate horizontal gene transfer of both resistance and virulence determinants, thereby increasing genetic variability and contributing to the spread of antimicrobial resistance within bacterial populations. The resistance of biofilms to conventional antibiotics is primarily driven by a combination of structural, physiological, and genetic factors, making the treatment of biofilm-related infections particularly difficult [38,59].

Accumulating evidence indicates that OMVs are actively integrated across the biofilm life cycle rather than representing inert by-products (Figure 1). At the attachment stage, OMVs modulate physicochemical interactions and surface hydrophobicity, thereby influencing initial cell–surface contacts [60,61]. During irreversible attachment, OMVs associate with eDNA and adhesin proteins, contributing structural support to the nascent matrix [51,52]. In the growth and maturation phases, OMV-associated proteins, lipids and signaling molecules (including QS-related factors) participate in biomass expansion and matrix remodeling, reinforcing community architecture [53,62,63]. Finally, during dispersal, OMVs are often enriched in degradative enzymes (proteases, lipases, nucleases) and can promote matrix breakdown, releasing previously sessile bacteria into a planktonic state to colonize new surfaces [55,57]. Together, these findings portray OMVs as dynamic regulators that balance biofilm stability and timely dispersal.

### 2.2. Biofilm Formation in A. baumannii

Numerous isolates of *A. baumannii* can establish biofilms on a wide range of surfaces, including both living tissues and inert materials, as demonstrated in vitro and supported by in vivo models and clinical isolates [64,65,66,67,68]. This phenomenon contributes to chronic and persistent infections, exacerbates disease processes, and is involved in antibiotic resistance confirmed in patient-derived isolates [64,69,70,71]. In *A. baumannii*, biofilm-associated virulence determinants, including specific genes and proteins, promote attachment to surfaces and contribute to biofilm development [64,72]. Key factors include bacterial fimbriae, pili and surface-adhesion proteins such as biofilm-associated proteins (Baps), which plays a critical role in biofilm initiation and maturation [64,72,73].

Key contributors to biofilm formation include cell-surface hydrophobicity, electrostatic properties, adhesion-related proteins, and extracellular polymeric substances [64,74,75,76]. Environmental changes in both physical and chemical settings impact phenotypic characteristics and the expression of key functions [64,77]. Biofilm growth is intricately regulated by several environmental factors at each stage, including temperature, osmolarity, ferrous iron concentration, nutrient availability, substrate quality, light exposure, oxygen levels, and ambient acidity [64,78]. Glucose is pivotal in *A*. *baumannii* biofilm formation [64].

Thus, it has been demonstrated that low iron availability not only slows the growth of *A. baumannii* but also impairs motility via type IV pili (T4P) and type I pili [72]. T4P are helical structures composed of protein subunits known as pilins. Their stability depends on hydrophobic interactions between N-terminal α-helices positioned within the core of the pilus fiber [79,80,81]. T4P are crucial for effective biofilm formation in vertical biofilm assays, as shown in vitro on abiotic substrates [79,82]. Type I pili, along with other pili, contribute not only to biofilm and pellicle formation in vitro [83,84] but also to the host immune response in cell culture models [83].

Additionally, a comprehensive transcriptome analysis comparing messenger ribonucleic acid (mRNA) [85] expression profiles of biofilm and planktonic cells in vitro has revealed a downregulation of genes related to motility and an upregulation of genes involved in biofilm formation. These genes also play a crucial role in attachment and adherence, which are the initial steps in establishing a mature biofilm [72].

A distinct type of pilus in *A. baumannii*, termed the CsuA/BABCDE-mediated pilus, has been described as the product of a chaperone-usher pathway. This pilus is prevalent in the majority of *Acinetobacter baumannii* clinical isolates, reflecting its conserved nature [64,67,72,86,87]. Structurally, it is organized into two main components: a slender tip fibrillum encoded by the *csuE* gene and a helical rod encoded by the *csuA/B* gene [72,88]. These subunits act, respectively, as a surface-recognition adhesion factor and as a structural element required for pilus biogenesis, with their expression strongly influenced by environmental conditions [72,89]. Experimental work has demonstrated that loss of CsuA/BABCDE pili production leads to impaired surface adhesion and a reduced ability to form biofilms on plastic substrates [72,90].

Subsequent studies further showed that while this pilus does not play a role in *A. baumannii* adherence to host cells in vitro (e.g., bronchial epithelial cells), it is indispensable for biofilm development on abiotic surfaces. Thus, the mechanism of host–cell attachment is independent of the CsuA/BABCDE-mediated pilus, which primarily supports abiotic surface colonization [64,72,91].

Two-component systems (TCSs), such as AdeRS, BaeSR, GacSA, OmpR-EnvZ, and PmrAB, primarily contribute to the virulence of *A. baumannii*. Additionally, AdeRS and BaeSR regulate drug efflux and mediate resistance to aminoglycosides, tetracyclines, tigecycline, erythromycin, fluoroquinolones, and chloramphenicol [72,92]. However, the BfmRS TCS, studied both in vivo and in vitro, governs the expression of the Csu pili chaperone–usher pathway, thereby affecting cell shape, promoting biofilm development, mediating adhesion to biotic as well as abiotic surfaces, and contributing to AMR [64,72,93,94]. Disruption of BfmRS inhibits host cell colonization and subsequent infection in animal models [72,94,95]. Moreover, BfmRS also regulates OMV production in *A. baumannii*, with BfmS acting as a negative regulator, as demonstrated in vitro, influencing both antimicrobial resistance and OMV-induced toxic effects on host cells. Consequently, BfmS appears to attenuate virulence-associated properties by modulating the bacterial cell envelope and OMV biogenesis [94].

A recently discovered TCS named CheAY also influences bacterial virulence by decreasing the expression of homoserine lactone synthase. This reduction impairs bacterial motility and biofilm formation in vitro, which are regulated by the CsuA/BABCDE chaperone/usher pili in an AbaI-dependent manner [72,96]. AbaI, an autoinducer synthase, produces signaling molecules essential for quorum QS. It plays a crucial role in enhancing biofilm formation in vitro and supporting normal biofilm development in *A. baumannii* [95,97,98].

Thus, studies in vitro demonstrate the archetypic QS system in *A. baumannii* is mainly composed of two essential proteins: the autoinducer enzyme synthase AbaI and the receptor-activating protein AbaR, along with N-acyl-homoserine lactones (AHLs) [73,99,100]. This system regulates gene expression in response to cell density through autoinducer signaling [101]. AHLs, synthesized by the AbaI enzyme, play a key role in QS, biofilm formation, drug resistance regulation, efflux pump activation, and motility as confirmed in vitro and in clinical isolates [73]. The *abaI* gene encodes AbaI, which catalyzes AHL production [102,103]. Additionally, AHL synthesis is dependent on the *luxI* gene, while AbaR is encoded by the *luxR* gene [100].

Baps have been identified in *A. baumannii* [73,104] as playing crucial roles in biofilm formation on abiotic surfaces in vitro such as polypropylene and titanium [73,105]. These proteins are involved in both biofilm maturation and intercellular adhesion, including bacterial adherence to human bronchial cells and neonatal keratinocytes in cell culture models [73,104]. Additionally, they help maintain cell surface hydrophobicity [73,105]. Bacteria that express the *bap* gene tend to produce more robust biofilms [73,106]. The *bap* gene is frequently found in multi-drug resistant (MDR) clinical *A. baumannii* isolates, indicating that this gene plays a role in the coordination of biofilm formation and reduced susceptibility to antimicrobials in *A. baumannii* [73,107].

Multiple genes participate in biofilm formation in vitro and in clinical isolates of *A. baumannii*, including *bla_PER_*_–*1*_, *epsA*, and *pt* [73]. The *bla_PER_*_–*1*_ gene located within the composite transposon Tn1213 [68,100,107] encodes a broad-spectrum β-lactamase known as *P*. *aeruginosa* extended resistance β-lactamase (PER-1) [41]. Its expression in *A. baumannii* is associated with poor clinical outcomes, as it is essential for MDR *A. baumannii* adherence to respiratory epithelial cells and for promoting biofilm formation on non-living surfaces [73,100].

Capsule production is a well-known virulence factor in *A. baumannii*. In vitro studies have shown that two genes, *epsA* and *ptk*, are involved in capsule biosynthesis [73]. The *epsA* gene encodes a predicted outer membrane protein, EpsA, which is classified as an exopolysaccharide [73,100,108]. This protein forms a structure at the cell surface, providing protection against external factors [73,100,108]. The *ptk* gene codes for a predicted protein tyrosine kinase (PTK) that synthesizes a polysaccharide capsule acting as a virulence determinant [73,100,109]. These two genes are associated with biofilm formation in MDR *A. baumannii* clinical isolates, supporting their potential to generate strong biofilms [73].

Other genes that significantly impact biofilm formation include *bla_VIM_*, *bla_OXA-23_* [73], *bla_OXA-51_*-like, and *ISAba1*, which influence both biofilm formation and antibiotic resistance patterns in clinical isolates, confirmed by in vitro assays [110].

OmpA (also referred to as AbOmpA) is a prominent porin located in the outer membrane of *A. baumannii* [73]. It has several pathogenic functions: promotes adhesion to and invasion of epithelial host cells [73,111,112], causes mitochondrial damage, and can induce cell death, as shown in vitro and in vivo [73,113,114]. AbOmpA supports biofilm growth on abiotic surfaces such as polystyrene and is also essential for firm attachment to biotic surfaces, as demonstrated in vitro using human alveolar epithelial cell lines [73]. This virulence factor is encoded by the *ompA* gene [64,115].

Mutations in the *bfmS* gene, which encodes the cytoplasmic membrane sensor kinase BfmS as part of the TCS BfmRS, can disrupt AbOmpA regulatory pathways and affect virulence [73]. In addition to AbOmpA, other proteins demonstrated in vitro to contribute to biofilm formation include the porin CarO together with the DNA repair and homologous recombination protein RecA [64].

In biofilms, the EPS matrix forms the main structural framework. It consists of polysaccharides, extracellular DNA (eDNA) originating from bacterial genomic material, lipids, and proteins [72,116]. eDNA is released through three pathways: active secretion, incorporation into membrane vesicles during early growth, or passive release following cell lysis in later stages. Its role in biofilm biomass has been demonstrated in vitro as eDNA presence increases biomass in culture, whereas exposure to DNase I leads to a marked reduction in biofilm mass. Although eDNA is a crucial component of biofilm formation, its exact role in the regulation of this process remains unclear [72,117]. In *P. aeruginosa*, eDNA serves as a structural scaffold, acts as an intracellular bridge, and functions as a molecular linker owing to its polyionic characteristics. Similar functions may be present in *A. baumannii*, but further research is needed to fully understand the role of various biofilm matrix components [72,118,119].

One of the best-characterized polysaccharides made by *A. baumannii* is poly-β-(1-6)-N-acetylglucosamine (PNAG), encoded within the *pgaABCD* gene locus [72,120,121]. PNAG plays a crucial role in structuring the biofilm, facilitating bacterial attachment, surface-to-cell adhesion, cell-to-cell interactions, and pellicle formation [72,120,121]. In vitro studies showed that the deletion of the *pgaABCD* gene cluster results in bacteria that aggregate but fail to form complex, multilayered biofilm structures. Additionally, increased expression of PNAG and its transporter has been reported in a colistin-resistant clinical isolates, indicating possible control by the two-component system PmrAB [72]. PNAG also contributes to bacterial protection against innate host defenses [64].

Biofilm growth and maturation are influenced by cyclic di-guanosine monophosphate (c-di-GMP), a cyclic dinucleotide detected in *A. baumannii*, which modulates both biofilm formation and movement across surfaces [72,122].

The biofilm matrix enveloping bacteria protects them from harsh environments and reduces susceptibility to antibacterial agents [64]. There is a positive correlation between biofilm formation and multidrug resistance in vitro and clinical settings [64,123]. Strains that form biofilms display heightened resistance to antibiotics such as ampicillin-sulbactam, amikacin, ciprofloxacin, and ceftazidime, while showing comparatively less resistance to imipenem and piperacillin [64,124].

Interestingly, data ICUs departments show variability, with some strong biofilm producers being more susceptible to antibacterial agents. This is because bacteria within biofilms do not always rely on the same resistance mechanisms as planktonic cells [64,125]. Some studies using in vitro biofilm assays with clinical isolates have indicated that *A. baumannii* strains capable of forming biofilms exhibit lower resistance rates to imipenem and ciprofloxacin compared with non-biofilm-forming strains [64,126]. In addition, *A. baumannii* strains resistant to meropenem tend to form biofilms less efficiently than meropenem-susceptible strains [64,127].

### 2.3. Biofilm Formation in P. aeruginosa

In the first stage of biofilm formation, *P. aeruginosa* attaches to the substratum using its extracellular appendages [128,129,130]. The flagellum of *P. aeruginosa* studied in vitro is essential for initiating biofilm formation due to its role in swarming and twitching motility [131,132]. After surface attachment, *P. aeruginosa* often downregulates or loses its flagellum, a phenomenon observed in clinical isolates from the mucus of cystic fibrosis patients, which helps the bacterium evade host immune responses and phagocytosis [131,133]. Once temporary attachment is established, *P. aeruginosa* strains extend their T4P, which promote stable cell-to-surface binding that becomes irreversible, as demonstrated in vitro and in computational models [130,134]. c-di-GMP plays a central role in controlling biofilm development, motility, adhesion, virulence, and morphological changes in *P. aeruginosa.* This has been demonstrated in vitro through genetic and biochemical studies [130,135]. Structural analyses and functional studies have indicated that elevated cellular concentrations of c-di-GMP enhance the production of matrix components, including exopolysaccharides, whereas basal levels promote dispersal and low concentrations correspond to the planktonic state. Thus, c-di-GMP regulates crucial processes contributing to the formation and maturation of *P. aeruginosa* biofilms [130,131].

Biofilm formation in *P. aeruginosa* is a multifaceted process, relying on the production of a self-generated EPS matrix composed of exopolysaccharides, eDNA, proteins, and lipids [131,136,137,138]. This matrix makes up over ninety percent of the biomass within the biofilm and serves as a structural framework that enables adhesion to both biotic and abiotic surfaces [136,138]. It also shelters embedded bacteria from unfavorable environmental factors, including antibiotics and host immune defenses, and supplies nutrients, enzymes, and cytosolic proteins critical for biofilm survival. Additionally, the extracellular framework supports intercellular communication [136,138].

Major extracellular polysaccharides produced by *P. aeruginosa* include alginate, pellicle (Pel) polysaccharide, and components of the polysaccharide synthesis locus (Psl), all of which contribute to surface attachment and to maintaining biofilm architecture [130,131,136,139,140,141]. Psl, a neutral pentasaccharide [130,136], facilitates cell attachment to surfaces and initiates biofilm development in both nonmucoid and mucoid *P. aeruginosa* strains, as demonstrated in vitro and confirmed in cystic fibrosis clinical isolates [136,139,142,143].

T4P are concentrated in Psl-rich regions, and greater local amounts of Psl promote short-range, localized attachment by aligning rod-shaped *P. aeruginosa* along the substrate [130,144,145]. Psl has several characteristics: (i) it benefits biofilm communities but not planktonic populations; (ii) Psl-deficient cells grow more effectively in heterogeneous biofilms when coexisting with Psl-producing bacteria, as the latter create conditions that support their growth [136,140,146]; (iii) in the process of biofilm development, Psl-positive strains surpass strains lacking Psl production; and (iv) Psl-deficient strains cannot exploit Psl producers [136,146]. In mature biofilms, Psl is positioned at the outer regions of mushroom-like structures, helping to maintain their structural integrity [136,142]. Psl as well serves as a signaling molecule, enhancing c-di-GMP synthesis; when this molecule’s level rises, the outcome leads to a denser and more resilient biofilm [136,146]. In addition, it shields biofilm-associated bacteria from antimicrobial agents and neutrophil-mediated phagocytosis, thereby serving as an important defense factor for sustaining chronic infection [136,147].

Pel polysaccharide is a positively charged polymer composed of incompletely deacetylated N-acetyl-D-glucosamine and N-acetyl-D-galactosamine. It is an essential component of biofilms formed by nonmucoid *P. aeruginosa* strains, playing a crucial role in initiating surface attachment and maintaining biofilm integrity [136,148,149]. However, Pel provides a transient adhesive force that is not as permanent as that of Psl [130].

Alginate is a capsular polysaccharide [130]. This acetylated polymer of high molecular weight, which is also an anionic polysaccharide, consists of β-1-4 glycosidic linkages between α-L-guluronic acid and β-D-mannuronic acid [130,150] and is mainly synthesized within the biofilm of mucoid *Pseudomonas* strains as a result of mutations in the mucA22 allele [136,151,152]. Strong evidence from in vitro studies, clinical cystic fibrosis isolates, and in vivo infection models demonstrates that overproduction of alginate by *P. aeruginosa* enables the bacterium to counteract inflammatory responses, neutralize host-derived reactive oxygen species, and evade clearance through phagocytosis by activated macrophages [130,153,154,155].

Moreover, overproduction of alginate by *P. aeruginosa* can suppress the synthesis of certain virulence determinants, including siderophores and rhamnolipids, during infection. Therefore, the host’s immune response is weakened, facilitating coinfections with other bacteria, including *S. aureus* [130,153,154,155]. In conclusion, alginate is an important factor in the progression of biofilm development, offering defense against uptake by phagocytes and opsonin-mediated recognition, and limiting antibiotic penetration across the biofilm structure [130,136].

Similar to *A. baumannii*, in vitro studies have shown that eDNA is an important factor [130,136] in the early stages of biofilm development, where it supports motility-driven expansion and is released into the environment through cell lysis [130,156,157]. When interacting with Pel and Psl, eDNA contributes in vitro to maintaining biofilm stability, whereas biofilms deficient in eDNA exhibit increased susceptibility to detergents [130,136].

Cell lysis may be induced by environmental stressors, including antimicrobial exposure, via the endolytic function of endolysin Lys encoded within the R- and F-pyocin gene clusters. Such lysis may take place during both the initial phases of biofilm formation and the planktonic state, when rod-shaped bacteria quickly shift into rounded cells because of cell wall damage, ultimately leading to lysis [136,156].

In this process, eDNA, cytosolic proteins, and RNA are later enclosed within membrane vesicles formed from fragments of membranes of lysed cells [136,156]; eDNA can additionally be present along the outer layer and within the stem-like portions of microcolonies exhibiting a mushroom-shaped architecture [136,158,159].

The extracellular DNA performs several functions: (i) serving as a nutrient store for bacterial cells embedded in the biofilm; (ii) assisting in the spatial arrangement and coordination of cells via twitching motility; (iii) binding divalent cations (Mg^2+^ and Ca^2+^) at the outer membrane, which subsequently initiates activation of the type VI secretion system that delivers virulence factors into the host; (iv) acidifying the biofilm environment and infection sites, thereby hindering the penetration of antimicrobial agents, as this pH change upregulates genes controlled by the PhoPQ and PmrAB TCSs, resulting in a marked increase in aminoglycoside resistance [131,136,160,161,162]; and (v) influencing the inflammatory response mediated by neutrophils in *P. aeruginosa* biofilms [136,163].

In vitro, the protein CdrA functions as an external adhesin that is released in association with the CdrA–CdrB system [130,164]. Its expression is regulated by c-di-GMP, with lower intracellular levels promoting greater CdrA export [130]. By binding to Pel and Psl, CdrA helps preserve biofilm structural integrity [130,165]. It has been established that even without exopolysaccharides, CdrA can promote cell clustering [130].

Laboratory studies have shown that intracellular c-di-GMP levels are regulated by the Wsp chemosensory-like signal transduction pathway, which is activated through the methyl-accepting chemotaxis protein WspA. In *P. aeruginosa*, WspR modulates the activity of c-di-GMP-binding receptors, with FleQ and PelD being key regulators of biofilm formation. FleQ promotes exopolysaccharide production by activating transcription of the corresponding operon, while simultaneously repressing flagellar gene expression, thereby suppressing surface sensing and favoring irreversible attachment [130,144].

An intracellular compound known as polyhydroxyalkanoate (PHA) was identified in *P. aeruginosa* biofilms using in vitro biofilm cultivation assays. Although PHA is synthesized within the bacterial cells residing in the biofilm, it does not directly contribute to the biofilm’s structural matrix. This polymer, recognized for its role in carbon and energy storage, is associated with increased bacterial resistance to environmental stress and supports the adhesion of cells to non-living surfaces, such as glass [136,166]. Under low-oxygen or anaerobic conditions within the biofilm, PHA is thought to function as an electron reservoir that facilitates energy metabolism [136,167,168].

Several key factors contributing to biofilm formation in *P. aeruginosa* include the QS systems LasI-LasR, RhlI-RhlR, PQS-MvfR, as well as the TCSs GacS/GacA and RetS/LadS [131,169,170,171]. Evidence from in vitro studies shows that QS regulates biofilm maturation, while in vivo infection models demonstrate its role in inflammation and persistence.

Evidence from in vitro studies shows that these QS systems are essential for the development of mature, structured biofilms [131], whereas findings from in vivo infection models highlight their importance in regulating acute and chronic processes that contribute to severe systemic infections [172,173]. For instance, MvfR plays a key role in intestinal inflammation during infection, while *P*. *aeruginosa*-induced intestinal permeability in murine models is regulated by a complex QS network involving MvfR, LasR, and RhlR [173]. The TCS GacS/GacA is a crucial signal transduction mechanism that regulates cellular communication and secondary metabolism [173,174]. In this system, the response protein GacA controls multiple downstream pathways, including the synthesis of QS signaling molecules AHLs and secondary metabolites. These pathways contribute to biofilm formation, motility, antibiotic resistance, and the progression of both acute and chronic infections [173,175,176,177]. The GacS histidine kinase serves as the membrane sensor in the Gac/Rsm regulon, along with its primary extracellular sensor, GacS/GacA. This pathway regulates the expression of numerous genes in pseudomonads, influencing bacterial fitness, motility, stress tolerance, biofilm formation, and virulence [175]. Thus, a *P. aeruginosa* strain deficient in GacA shows a tenfold decrease in biofilm-forming ability compared to its wild-type counterpart PA14, indicating that the GacS/GacA system positively regulates biofilm formation [131,178]. Two additional histidine kinases, RetS and LadS, have been identified as components of the Gac system [179,180]. In vitro studies indicate that RetS represses genes essential for biofilm formation [180,181], whereas LadS activates genes that promote biofilm development [131]. Consequently, the RetS and LadS regulons contribute to the synthesis of key EPSs, such as Pel and Psl, which are integral to the biofilm matrix [182,183,184]. Moreover, the RetS and LadS regulons in *P. aeruginosa* regulate a broad range of virulence factors, including genes involved in motility [182,183,185]. Additionally, these regulons control the expression of multiple secretion systems [182].

Under specific conditions, *P. aeruginosa* can undergo significant physiological and phenotypic changes during biofilm formation. For instance, during chronic cystic fibrosis infections, the bacterium can adopt a mucoid phenotype characterized by increased alginate production and biofilm colony formation. This adaptation has been consistently observed in clinical isolates from cystic fibrosis patients [131]. Additionally, *P. aeruginosa* secretes cyclic glycerophosphorylated β-(1,3)-glucans into the extracellular biofilm matrix, which can sequester kanamycin [131,186]. Functional genomics in vitro using the *P. aeruginosa* clinical isolate PA14 has identified several genes, including *ndvB*, *PA14_40260–40230*, and *tssC1*, which influence biofilm-associated antibiotic resistance rather than biofilm formation itself [131,187,188,189]. The *ndvB* gene encodes a glucosyltransferase that synthesizes cyclic β-(1,3)-glucans, which bind tobramycin and sequester it in the periplasm [131,187]. The *PA14_40260-40230* operon encodes a novel efflux pump, and its deletion reduces resistance to gentamicin and ciprofloxacin in biofilms [131,189]. The *tssC1* gene, which encodes a component of the type VI secretion system and is strongly upregulated within biofilms, contributes to resistance against tobramycin, gentamicin, and ciprofloxacin [131].

### 2.4. Biofilm Formation in K. pneumoniae

The capacity of *K. pneumoniae* to establish biofilms significantly contributes to its role in hospital-acquired infections. Most clinically isolated *K*. *pneumoniae* strains have been reported to form biofilms. Strains with strong biofilm formation are often retrieved from clinical materials such as urine, pus, and blood, whereas those with weaker biofilm activity are more commonly found in blood specimens [190].

Fimbrial structures facilitate the attachment of *K. pneumoniae* cells to inert surfaces, a key factor in the development of infections associated with medical devices like catheters [190]. Genomic analyses have revealed that this bacterium harbors at least ten gene clusters encoding chaperones, ushers, and adhesin proteins involved in fimbrial assembly, including the *fim*, *mrk*, *ecp*, and *kpa–kpg* clusters [36,191,192,193]. Among these, *fim*, *mrk*, *ecp*, and *kpf* are the most extensively characterized, encoding type I and type III fimbriae [36,194], the common pilus [36,192], and type I-like fimbriae [36,195].

However, the *fim* and *mrk* gene clusters, which direct the production of type I and III fimbriae, have been most extensively characterized in biofilm-related studies, including in vitro assays and in vivo studies using murine models. In *K. pneumoniae*, type III fimbriae are considered key contributors to the initial stages of biofilm development, with their components synthesized under the control of the *mrkABCDF* operon. These structures are largely composed of the MrkA protein, which assembles into a coiled filament crucial for stable adherence to abiotic surfaces in vitro [190].

MrkA protein expression is significantly upregulated during biofilm thickening. MrkB and MrkC proteins have sequence characteristics representing the periplasmic chaperone and usher translocatase, respectively [190]. MrkD, located at the tip of the fimbriae, supports adhesion and helps define the selectivity of surface recognition and attachment [190,196]. The *mrkA* and *mrkD* genes are critically involved in regulating biofilm development processes in *K. pneumoniae* [190,197]. In vitro, the *mrkA* gene plays a major role in accelerating biofilm development, whereas *mrkD* contributes to enhancing the compact architecture of biofilms formed by *K. pneumoniae* [190,198]. The regulatory cluster *mrkHI* is known to influence the expression dynamics of type III fimbriae. *MrkH* functions as a recently identified transcriptional regulator of the mrk cluster, modulating *mrkHI* activity and containing a PilZ domain [190,199,200]. This regulator binds to the region upstream of the *mrkA* promoter and enhances transcription of the downstream *mrkABCDF* operon. For this reason, MrkH is described as a “biofilm switch,” as it activates genes required for the assembly of type III fimbriae in vitro [190,199,200].

In vitro assays using mutant strains have shown that transcription of *mrkHI* can also be triggered by MrkI, a LuxR-type regulatory protein [190,199,201]. Additionally, its expression is modulated by the ferric uptake regulator Fur, acting at the transcriptional level to promote *mrkHI* expression. When Fur is disrupted, expression of *mrkH*, *mrkI*, and *mrkA* is diminished, leading to reduced synthesis of type III fimbriae and weaker biofilm formation. Attachment mediated by these fimbriae is thought to be one of the earliest steps in *K. pneumoniae* adherence to abiotic surfaces [190,202]. Moreover, type III fimbriae are essential for regulating the transition from a motile to sessile bacterial state in response to c-di-GMP, the principal intracellular signaling molecule involved in biofilm-related functions. This process involves several genes, including the *mrkHIJ* gene cluster, *MrkJ* [190], and *YjcC* [190,203].

The transcriptional regulator IscR negatively modulates type III fimbriae production in *K. pneumoniae*, and its deletion has been associated with increased biofilm-forming potential [36,204]. This repression occurs as IscR directly inhibits the synthesis of the regulators MrkH and MrkI [36].

Type I fimbriae, which are sensitive to mannose, consist of the primary structural protein FimA and the distal adhesin FimH. These components are encoded by the *fimAICDEFGHK* gene cluster [190,195]. In vitro, their expression is governed by the invertible DNA segment (*fimS)*, which acts as a reversible genetic switch [190,192]. Elevated levels of *fimH* expression enhance bacterial surface binding and contribute to robust biofilm architecture, as shown in vitro and in clinical isolates [190,198]. Within the *fim* operon resides the regulatory gene *fimK*, which encodes the FimK protein containing an EAL domain—a conserved motif responsible for c-di-GMP-specific phosphodiesterase activity and named after its Glu-Ala-Leu sequence. This domain allows FimK to modulate intracellular c-di-GMP levels. In vitro, FimK downregulates type I fimbriae production, thereby limiting biofilm formation and reducing bacterial aggregation in host tissues [190].

KpfR, a known negative regulator of type I fimbriae, has been shown in vitro to restrict the development of multicellular communities in *K. pneumoniae*. Consequently, deletion of either *fimK* or *kpfR* results in bacteria with intensified fimbrial expression and a greater capacity for biofilm development. The *kpfR* gene belongs to the *kpf* locus, which includes *kpfR*, *kpfA*, *kpfB*, *kpfC*, and *kpfD*, and is associated with the biosynthesis of fimbriae structurally similar to type I [36].

In addition to the *kpf* cluster, several newly identified fimbrial gene clusters, including *kpa*, *kpb*, *kpc*, *kpd*, *kpe*, and *kpg*, have been described in *K. pneumoniae* [190,191]. The *kpgC* gene plays an important role in enhancing biofilm growth, mediating bacterial adherence to host cells, and enabling colonization of the intestinal tract in murine hosts. Loss of *kpgC* disrupts these essential processes. Similarly, *kpaC* and *kpeC* mutants exhibit reduced biofilm formation, with *kpaC* deletion affecting biofilm development at both early and late stages, while *kpeC* loss specifically diminishes adhesion to *Arabidopsis* cells. The *kpjC* gene, encoding a fimbrial usher, is also essential for biofilm formation, as its absence markedly reduces biofilm production [190,193].

Additionally, the *E. coli* common pilus (ECP) fimbriae gene cluster has been identified in the *K. pneumoniae* genome. This cluster includes the genes *ecpRABCDE*, and approximately 90% of *K. pneumoniae* strains are capable of producing ECP fimbriae [36,190].

In *K. pneumoniae*, ECP fimbriae promote bacterial aggregation and facilitate the formation of localized microcolonies on epithelial cell monolayers in vitro, contributing to persistent bacterial structures attached to abiotic surfaces. These structures are considered important for adhesion, community organization, and persistence in diverse environments—particularly in strains lacking the MrkD adhesin or type III fimbriae [190].

In conclusion, type I fimbriae primarily contribute to urinary tract infections by binding to mannose residues on bladder epithelial cells, facilitating adhesion, invasion, and the formation of intracellular bacterial communities with biofilm-like properties. Both type I and type III fimbriae support biofilm formation and can compensate for each other; however, type III fimbriae play a more dominant role in this process. Notably, bacterial strains lacking type I fimbriae can form biofilms as effectively as those that express them [190].

*K*. *pneumoniae* exhibits a wide diversity of exopolysaccharide profiles, which correspond to variable capsular antigenic compositions. The polysaccharide capsule plays a multifaceted role in biofilm dynamics, from early adhesion stages to later biofilm maturation, as demonstrated in vitro using *K. pneumoniae* isolates [205,206]. The capsule is essential for constructing the initial covering of a mature biofilm structure [190]. When capsule synthesis is disrupted, bacteria often exhibit compromised biofilm-forming capacity [36]. In vitro, one known mechanism involves transposable element insertions within *ORF12* of the K2 capsular region, leading to reduced capsule biosynthesis and defective biofilm development on surfaces mimicking the human extracellular matrix [205].

Genes that contribute to capsule-dependent biofilm organization include ORF4, a *wza*-like gene implicated in polysaccharide translocation; ORF14, which encodes a glycosyltransferase participating in sugar assembly; *wzm*, associated with the transport of lipopolysaccharide (LPS) precursors; and *wbbM*, linked to LPS synthesis. Mutations in *wza* and ORF14 impair cell adhesion, while insertions into the *wza* or *wzc* regions further compromise capsular integrity and biofilm formation in vitro [205,207]. LPSs promote the initial attachment of *K. pneumoniae* to abiotic surfaces, essential for early biofilm formation. Mutants lacking genes involved in LPS biosynthesis (*wbbM*) or transport (*wzm*) exhibit delayed biofilm development. Both *wbbM* and *wzm* are upregulated in biofilm-grown *K. pneumoniae* compared to planktonic cells in the exponential phase [36,208].

Genes involved in the regulation of capsular carbohydrate pathways, including *treC*—which encodes an enzyme involved in the breakdown of trehalose-6-phosphate—and *sugE*, responsible for the production of an inner membrane-associated protein, influence biofilm-associated behaviors through their impact on capsule polysaccharide biosynthesis. In *K. pneumoniae*, such alterations can contribute to liver abscess formation. In vivo, *treC* mutants exhibit weakened gastrointestinal colonization ability, whereas *sugE* mutants show increased biofilm formation, higher mucus viscosity, and enhanced capsular polysaccharide production in vitro. The lack of *sugE* expression in *K. pneumoniae* has been associated with alterations in membrane integrity and the initiation of downstream regulatory pathways, ultimately leading to enhanced capsule biosynthesis during biofilm development [36,205,207].

The virulence gene *wcaG*, involved in capsule biosynthesis, has been shown in vitro to positively correlate with biofilm formation. Mutations in *wcaG*, which encodes a protein participating in the biosynthesis of fucose, lead to reduced biofilm formation by affecting capsule polysaccharide composition [190,209].

The virulence gene *wcaG*, involved in capsule biosynthesis, has been shown in vitro to positively correlate with biofilm formation. Inactivation of this gene significantly reduces capsule retention and increases biofilm formation [190,210].

The QS network contributes to the regulation of multiple surface-associated structures, including fimbriae, capsular components, and adhesive factors, by mediating intercellular signaling. This system plays a role in coordinating biofilm initiation, structural organization, and maturation within bacterial populations [190,210,211]. In *K. pneumoniae,* type II QS primarily relies on AI-2 signaling, which facilitates interspecific communication by allowing bacteria to respond to AI-2 produced by other species as well as their own. This has been demonstrated both in vitro and in clinical isolates [36,190,205,212,213]. The LuxS enzyme facilitates the generation of AI-2, a signal molecule derived from cyclic furanone-based structures, marking a key step in Type II QS. In *K*. *pneumoniae*, a gene related to *luxS* has been detected, supporting the presence of a communication system dependent on LuxS-mediated signaling [36,212].

*K. pneumoniae* strains lacking AI-2 signaling components show increased surface coverage following dynamic micro-fermentation growth but develop thinner biofilms [190,214]. In vitro, deletion of the *luxS* gene alters biofilm architecture, resulting in reduced surface coverage and fewer large colonies [205,212,213]. Mutations affecting *luxS* and AI-2 transport systems also upregulate *wbbM* and *wzm*, genes involved in LPS biosynthesis, suggesting that QS influences biofilm formation through LPS modulation in *K. pneumoniae* [36,190,205]. Notably, type II QS is associated with decreased *wzm* expression and increased expression of *pgaA*, which encodes a porin for PGAN [205,212,213].

Although *K. pneumoniae* does not produce AHLs, the signaling molecules of type I QS, it encodes SdiA, an orphan LuxR-type receptor that detects AHLs from other bacterial species in vitro. SdiA represses type I fimbriae expression in *K. pneumoniae*. Loss of SdiA results in a hyperfimbriated phenotype, enhancing biofilm formation and yeast cell agglutination in the *sdiA* mutant strain [36,190,205,215].

Efflux pumps contribute to various aspects of biofilm development. These roles include: (i) the removal of EPSs and QS signals, which supports the construction of the biofilm matrix; (ii) modulation of gene expression pathways related to biofilm development; (iii) extrusion of harmful agents such as antimicrobial drugs and metabolic by-products; and (iv) influencing bacterial attachment through mechanisms that either enhance or inhibit adherence to biotic and abiotic surfaces [205,216]. In *K. pneumoniae*, the efflux systems AcrA and OqxA are categorized under the Resistance-nodulation-cell division (RND) family, while QacEΔ1 (a truncated variant of QacE) and EmrB are classified as members of the Small multidrug resistance (SMR) family and the Major facilitator superfamily (MFS), respectively [205,217]. Transcriptomic analyses reveal that the expression levels of *acrA*, *emrB*, *oqxA*, and *qacEΔ1* are markedly higher in biofilm-embedded cells compared to cells growing in a free-floating, non-attached state [205,218].

Iron deprivation in *K*. *pneumoniae* is one of the signals that triggers infection and colonization. Thus, iron availability serves as a pivotal regulatory signal influencing a variety of virulence-associated functions, including the development of biofilms, as demonstrated in vitro [36,195]. It has been established that *K. pneumoniae* cells in biofilms, or those that have recently dispersed from a biofilm, may downregulate siderophore production to avoid stimulating a strong immune response and thereby evade the host’s immune system, as supported by transcriptomic studies of infected murine cells [36,219].

In *K. pneumoniae*, intracellular iron levels are tightly controlled by the global transcriptional regulator Fur [36,195], whose activity varies based on its iron-binding status—acting as either a repressor or an activator of target genes depending on whether iron is bound or not [36,195,220]. Beyond its role in iron metabolism, Fur serves as a global regulator influencing genes linked to virulence, including those involved in biofilm formation [36,221]. Under iron-rich conditions, Fur modulates biofilm development by regulating type I [36,195,211] and type III fimbrial gene expression [36,202], as well as transcriptional regulators of fimbriae [36,195,202].

### 2.5. Biofilm Formation in Enterobacter spp.

Within the *Enterobacter* genus, *E. aerogenes* and *E. cloacae* are among the most frequently encountered species in clinical settings and are often associated with hospital-acquired respiratory and urinary tract infections [222,223]. Despite their medical relevance, current understanding of QS mechanisms and virulence regulation in these bacteria remains limited [223,224,225].

As demonstrated in in vitro and in clinical isolates, curli fimbriae [223,226] and the second type VI secretion system (T6SS-2) [223,227] play crucial roles in bacterial adhesion to abiotic and biotic surfaces, as well as invasion. Furthermore, regulation of biofilm formation has been linked to the transcriptional activity of curli biosynthesis genes, including *csgA* and *csgD* [223,228], whose expression is essential for maintaining biofilm architecture and function [223].

QS in *Enterobacter* species is mediated through several classes of autoinducers, including AI-1, AI-2, and AI-3, which function as molecular signals regulating gene expression and collective behavior, as demonstrated in vitro using isolates obtained from plants, soil, and the human host [223,229]. In vitro, AI-1 signaling relies on acyl-homoserine lactones such as C4-HSL and C6-HSL [225,229], with regulatory input provided by LuxR-family proteins. These regulators have been associated with decreased bacterial adhesion and suppression of biofilm development [229,230]. AI-2 signaling contributes to intercellular coordination among *Enterobacter* spp., facilitated by Lsr-type receptors that have been identified through genomic and in vitro studies in *E. cancerogenus*, *E. cloacae*, and *E. mori* [223,231,232]. AI-3, in conjunction with host stress hormones such as epinephrine and norepinephrine, influences biofilm-related phenotypes through activation of the QseC/QseB two-component systems and has been demonstrated in vitro and in commensal *E. cloacae* clinical isolates [229,233].

Recent findings from in vitro functional studies, further supported by systematic reviews, indicate that the LuxR-type transcription factor EasR, identified in *E. asburiae*, is capable of sensing diverse AHL molecules and initiating transcription from QS-responsive promoters [223,229,234].

In vitro research by Liu et al. has demonstrated that *E. kobei*—a species classified within the *E. cloacae* complex—retains the ability to form robust biofilms under nutritional stress, yet exhibits comparatively lower resistance and attenuated pathogenic potential than other species within the same complex [235].

The summarized data on biofilm formation in ESKAPE Gram-negative bacteria is presented in Table 2.

## 3. OMVs in Gram-Negative ESKAPE Bacteria

### 3.1. Key Aspects of OMV Formation and the Functions of OMVs

OMVs are nanoscale, double-layered particles derived from the external membrane of Gram-negative bacterial species. Although both Gram-positive and Gram-negative microorganisms are capable of producing extracellular vesicles, the designation “OMVs” is typically reserved for those originating from Gram-negative bacteria due to their characteristic outer membrane structure (Figure 2) [34,38].

These vesicles arise from the bacterial cell envelope, which consists of an asymmetrical bilayer: the internal leaflet contains phospholipids (PLs), whereas the external leaflet is enriched with LPSs. LPS is composed of phosphorylated glucosamine residues connected to fatty acid chains that give rise to lipid A. Lipid A further binds to conserved core sugars and may be extended by a polymeric saccharide sequence known as the O-antigen [38,236].

Outer membrane-associated proteins are mainly divided into β-barrel structures that traverse the membrane and lipidated outer membrane lipoproteins. Some of these proteins, including outer membrane proteins (OMPs) and lipoproteins, interact with the peptidoglycan (PG) layer through non-covalent interactions, contributing to membrane stability. Moreover, the lipoprotein Lpp anchors the outer membrane to the PG layer, supporting mechanical cohesion and contributing to the structural robustness of the bacterial surface architecture through covalent linkages [38,236].

The formation of OMVs is governed by numerous environmental and physiological variables [38,237]. Exposure to specific antimicrobial agents has been shown to upregulate OMV production. For example, fluoroquinolones and β-lactams such as ciprofloxacin, meropenem, fosfomycin, and polymyxin B increase vesicle release in *E. coli*, while carbapenem exposure intensifies vesiculation in multidrug-resistant *K. pneumoniae*. OMV emergence initiates through membrane curvature and outward deformation, culminating in vesicle detachment from the bacterial surface [34,38].

OMV formation is commonly described by two mechanisms: lytic (occurring during cell lysis) and non-lytic (involving outer membrane blebbing) [38,236]. Lytic OMV-mediated activity involves PG-derived fragments and vesicle-associated hydrolytic enzymes with potential autolytic function. In this model, an uneven spatial distribution of PG precursors during biosynthesis may induce localized outward deformation of the external membrane, initiating a signaling cascade that culminates in vesicle emergence. Experimental data suggest that alterations in autolytic enzymes participating in peptidoglycan remodeling are associated with elevated production of OMVs. This correlation points to a possible role of accumulated cell wall-derived components in generating outward mechanical force on the outer layer, facilitating vesicle detachment [38,237].

The non-lytic pathway, in contrast, involves multiple mechanisms: weakening of outer membrane–cell wall anchoring points, enhanced curvature of the outer surface, build-up of periplasmic pressure, and mobilization of flagellar elements [38,236]. The connection between the external membrane and the PG framework is largely maintained by the lipoprotein Lpp, which exists in both bound and unbound configurations. LdtF (also referred to as DpaA), a protein identified in *E. coli*, has been shown to enzymatically cleave Lpp–PG linkages. Impaired binding between Lpp and the cell wall PG—whether due to genetic deletion or reduced interaction efficiency—has been associated with elevated vesicle generation. Slight weakening of the Lpp–PG association has been shown to promote vesicle release [38,236]. When only a few lipoproteins interact with the PG layer, localized deformation of the outer membrane may occur, facilitating vesicle emergence [38,237,238].

A second pathway contributing to OMV biogenesis involves structural changes in the outer membrane lipid bilayer. Alterations in LPS composition, including mutations in genes involved in its biosynthesis or enzymatic removal of acyl chains, have been associated with enhanced vesicle release. Deacylated LPS molecules containing five acyl chains can promote membrane pliability and support OMV formation; nevertheless, strains relying solely on such structures often display reduced vesicle output. Redistribution of phospholipids within the outer membrane—particularly their enrichment in the external leaflet, as observed in *H. influenzae* and *V. cholerae*—has been associated with increased vesicle formation. This phenomenon may be driven by reduced membrane rigidity and resembles the membrane destabilization seen after specific outer membrane protein loss or exposure to chelating agents such as ethylenediaminetetraacetic acid (EDTA). In *N. meningitidis*, limited sulfur availability has been linked to increased phospholipid biosynthesis, a shift that supports OMV generation. The structural stability of the outer membrane relies in part on interactions between LPS and divalent ions such as calcium and magnesium. Disruption of these ionic bridges—such as through chelation by EDTA—has been shown to reduce membrane cohesion and promote vesicle release [38,236].

Molecules with negative surface charge, such as Pseudomonas Quinolone Signal (PQS), can reduce outer membrane stiffness. In *P. aeruginosa*, OMVs frequently contain negatively charged LPS variants, although even mutants lacking these structures continue producing vesicles with different biophysical characteristics. Shifts in temperature impact OMV generation differently across bacterial species. For instance, higher temperatures stimulate vesiculation in *E. coli*, while *P. aeruginosa* remains unaffected. Opposing responses have been noted in *S. marcescens* and *B. henselae*. Bacterial membrane fluidity appears to be modulated by temperature-driven changes in fatty acid saturation, supporting adaptation to thermal variation. In summary, vesiculation is a complex route that may involve other, less understood factors [38,236].

A third pathway contributing to OMV generation involves rising tension within the periplasm, driven by the intracellular accumulation of misfolded proteins and other periplasmic contents. In *P. aeruginosa*, impairments in outer membrane protein biogenesis intensify the concentration of unfolded components in the periplasm, which in turn facilitates vesicle release. Remarkably, this process may still occur in the absence of significant periplasmic load, provided that membrane-associated degradation enzymes are disrupted. Inactivation of periplasmic proteases in this species results in the accumulation of improperly folded proteins, a condition that enhances vesicle release through a process not reliant on PQS-mediated signaling. Under envelope stress linked to the buildup of non-native periplasmic proteins, specific small non-coding RNAs become transcriptionally active, leading to reduced OmpA expression and facilitating increased vesicle formation [38,236].

Finally, a fourth pathway implicates the bacterial flagellum in OMV release. In certain Gram-negative species, vesicle formation has been associated with the rotational movement of LPS-covered flagella. Vesicles linked to flagellar structures may possess distinct proteomic and lipidomic profiles compared to those derived from the canonical outer membrane. Overall, OMV formation occurs through diverse mechanisms that are influenced by bacterial physiological and ecological demands [38,236].

Upon release, OMVs encapsulate a complex mixture of components, including outer membrane-derived factors such as LPSs and OMPs, PG fragments, PLs, lipooligosaccharides (LOS), and various periplasmic constituents like hydrolytic enzymes and virulence-associated molecules. Nucleic acids, including DNA and RNA, have also been identified within these vesicles, although the precise pathways responsible for their incorporation and transport remain unclear. Variations in OMV molecular profiles are influenced by microbial identity, nutrient availability, and ambient environmental conditions. Proteinaceous load forms a substantial portion of OMV content and contributes to a wide range of molecular interactions and biological processes. Among them, porin-like molecules such as OmpF, OmpA, and OmpC facilitate interactions with host barriers, often mediating attachment via receptor-ligand binding mechanisms [38,237].

The adhesive capabilities of OMVs play a central role in microbial invasion and disease progression. These vesicles enable the transport of harmful bacterial components to remote target sites while shielding them from degradation under stressful chemical conditions. By doing so, OMVs reduce the need for direct bacterial contact with host cells. For instance, *P. aeruginosa* releases vesicles enriched with lytic factors that disrupt microbial membranes, exerting antimicrobial action against both Gram-negative and Gram-positive microorganisms. Moreover, vesicles derived from *P. aeruginosa* and *B. fragilis* may include specific surface-bound proteins—like aminopeptidase and hemagglutinin—that facilitate bacterial anchoring to host epithelial structures [38,236].

OMVs can alter immune responses through interactions with epithelial barriers. These vesicles enter host tissues via multiple uptake mechanisms, such as membrane fusion or endocytic internalization. Once inside the cells, OMVs stimulate inflammatory responses and elevate the production of pro-inflammatory cytokines. Molecular components packaged within OMVs—such as LPSs, flagellin, PG, Lpps, DNA, and RNA—act as pathogen-associated molecular patterns (PAMPs), recognized by host cell pattern recognition receptors (PRRs), thereby initiating immune responses. The specific PRR signaling pathways that are activated by OMVs can vary depending on the bacterial species involved [34,38].

The formation of OMVs serves as a mechanism to counteract envelope stress, helping bacteria survive under adverse conditions [38,239]. Recent studies have shown that elevated levels of AlgU (also known as σH), an alternative extracytoplasmic sigma factor functionally analogous to σE or RpoE in *E*. *coli*, are associated with increased OMV production in *P*. *aeruginosa*. On the other hand, the absence of AlgU leads to excessive vesiculation, highlighting the role of OMVs in relieving envelope stress [38,238,239].

Moreover, the formation of OMVs contributes to the management of oxidative damage. A well-documented example involves *P. aeruginosa*, where ciprofloxacin-induced DNA damage triggers the SOS response and increases OMV output [38,239]. The SOS regulatory system orchestrates diverse stress-associated processes, including modulation of cell cycle progression, environmental stress adaptation, mobilization of latent prophages, initiation of biofilm formation, expression of pathogenic determinants, and induction of low-fidelity DNA synthesis [240].

OMVs play a significant role in neutralizing antibiotics and antimicrobial agents. Some bacterial species release outer membrane vesicles capable of sequestering detergents or excess heme, thereby diminishing the local concentration of these damaging substances. This shielding capacity is primarily attributed to the outer membrane composition and can be transferred between microorganisms through OMVs [38,236].

Moreover, OMVs contribute to AMR propagation by serving as vehicles for horizontal gene transfers. This pathway serves as an alternative to classical gene transfer methods—such as transformation, transduction, and conjugation—by bypassing barriers like host range restrictions, limited DNA cargo size, and structural incompatibility of the transferred material. Genetic elements, including antimicrobial resistance genes (ARGs), may be packaged within OMVs and horizontally transmitted between bacterial cells. This ability enables OMVs to distribute resistance determinants within diverse bacterial populations, allowing bacteria to acquire resistance from even distantly related strains and promoting the widespread spread of resistance. This phenomenon has been documented across several Gram-negative microbial species, such as *A. baumannii*, *E. coli*, *P. gingivalis*, *N. gonorrhoeae*, and *P. aeruginosa*. Clarifying the role of OMVs in AMR is crucial, given their capacity to facilitate horizontal dissemination of resistance determinants and compromise infection control strategies [38,237].

Besides trapping bactericidal agents, these vesicles may transport enzymes involved in bacterial defense. Thus, OMVs originating from *A. baumannii*, *M. catarrhalis*, *E. coli*, and *S. maltophilia* have been reported to contain active β-lactamases [38,236,237]. These enzymes, when enclosed in OMVs, are shielded from antibody-mediated inhibition, thereby preserving their functionality and enhancing bacterial tolerance to β-lactam antibiotics [38,236,237]. Notably, OMVs from *S. maltophilia* not only provide protection to the originating strain but may also transfer this enzymatic activity to neighboring bacterial populations, such as *P. aeruginosa* and *B. cenocepacia.* Likewise, OMVs carrying β-lactamases from *M. catarrhalis* have been associated with improved persistence of *H. influenzae* and *S. pneumoniae* [34,38,237]. Additionally, vesicles released from β-lactam-resistant *E. coli* can degrade these antibiotics in a concentration-dependent manner, thereby protecting β-lactam-susceptible *E. coli* strains from growth inhibition [34,38]. The antimicrobial defense capabilities of OMVs extend beyond β-lactam inactivation. They are also implicated in neutralizing other antibiotics, such as colistin, melittin, and polymyxin B. Certain OMVs carry catalase—an enzyme with antioxidant activity—that helps neutralize oxidative stress in bacterial cells. Others may deliver proteolytic enzymes that cleave host immune-related proteins and signaling molecules [38,236]. Moreover, OMVs contribute to the establishment and architecture of biofilms by transporting regulatory factors that influence extracellular matrix production and microbial spatial organization. *P. aeruginosa* OMVs, for instance, can carry PQS, a signaling molecule that modulates biofilm composition and facilitates interspecies communication [38,237]. As discussed in Section 2.1, the interplay between OMVs and biofilms is bidirectional, with vesicles supporting adhesion, matrix stabilization, and dispersal, while the biofilm lifestyle itself modulates OMV composition and release.

It has been recently discovered that OMVs in tumors, including gastric, colon, oral, and lung cancer, contribute to carcinogenesis and progression through various mechanisms. However, their precise roles within the tumor microenvironment remain poorly understood, limiting a full exploration of their relationship with tumors [241].

#### Biomedical and Clinical Applications of OMVs

OMVs are not only central to bacterial physiology and antimicrobial defense but are also increasingly recognized for their translational and therapeutic potential (Figure 3) [38]. In clinical practice, OMV-based vaccines have already achieved licensure, most notably the meningococcal B vaccine, which incorporates *Neisseria meningitidis*-derived OMVs and demonstrates safety and protective efficacy in humans [37,242]. Beyond this established example, preclinical research has explored OMVs as platforms for vaccines against other Gram-negative pathogens, including *Shigella*, *K. pneumoniae*, and *P. aeruginosa*, where protective effects were demonstrated in animal infection models [242]. The intrinsic immunostimulatory properties of OMV-associated proteins and PAMPs further enhance their utility as natural adjuvants in vaccine formulations [243].

Engineered OMVs are also being evaluated as drug delivery vehicles capable of transporting antimicrobial compounds, RNA-based therapeutics, or small molecules directly to target sites as well as agents that inhibit bacterial adhesion, with promising outcomes demonstrated in preclinical studies [243,244,245]. In oncology, OMVs engineered with tumor-associated antigens or immune checkpoint inhibitors have been shown to elicit anti-tumor immunity and to synergize with existing therapies, while also serving as carriers for chemotherapeutic drugs [246]. As an example, OMVs from *Moraxella catarrhalis* have been tested as carriers to overcome fluconazole resistance in yeast [244], while *K. pneumoniae*-derived OMVs have been shown to deliver doxorubicin in breast cancer models [247]. Furthermore, OMVs are being investigated as potential diagnostic biomarkers [248]. Together, these findings underscore the dual nature of OMVs: while originally viewed as bacterial stress-response structures, they are now being harnessed as versatile tools in vaccine development, drug delivery, immunotherapy, and diagnostics [38].

### 3.2. OMV Production in A. baumannii

OMVs released by *A. baumannii* span approximately 10–300 nm in size and are generated through outer membrane remodeling processes during bacterial proliferation [249]. The outer membrane can bend and deform, forming vesicles due to increased curvature. This membrane deformation causes spherical vesicle formation. The aggregation of PL can change the curvature; silencing or deleting *VacJ/Yrb* genes results in increased PL accumulation and asymmetric outer membrane dilation. During PG layer synthesis, high PG concentrations cause outer membrane protrusions, indicating OMV formation. LPS is a major component of OMV formation in Gram-negative bacteria, with mutations in LPS-related genes affecting OMV biogenesis [250]. Interestingly, *A. baumannii* is capable of generating OMVs even in the absence of LPS, suggesting that LPS is not a critical component for their production [250].

The OmpA protein interacts with PG and the outer membrane, determining vesicle formation sites and promoting pathogenesis and dissemination by activating host cell processes [251,252]. OMVs enriched with OmpA are internalized by host cells, where they trigger the activation of the dynamin-related GTPase protein 1 (DRP1). This OmpA-driven stimulation promotes DRP1 accumulation on mitochondrial membranes, leading to mitochondrial disruption, elevated production of reactive oxygen species (ROS), and, ultimately, host cell death. Notably, suppression or loss of DRP1 can reverse these harmful cellular effects. Thus, OmpA and the OMVs that carry it contribute significantly to *A. baumannii* pathogenesis by serving as important virulence determinants [252].

Multiple molecules, such as LPS, OMPs, and PLs, contribute to vesicle formation [251]. High temperatures enhance membrane fluidity and OMV production, while pH fluctuations can affect OMV production via LPS modifications. OMVs naturally contain PLs, OMPs, LPSs, and soluble periplasmic proteins [249]. In *A*. *baumannii*, PLs and LPSs are vital for maintaining OMV morphology and stability. Microdomains enriched in sphingomyelin and cholesterol within the OMV membrane are thought to support interactions with host cells by promoting vesicle-mediated signaling. The PL bilayer forms the OMV walls, protecting the cargo and participating in bacterial activities like energy generation. OMPs are pivotal in cell invasion and apoptosis, aiding *A*. *baumannii*’s survival in the host [251].

OmpA mediates attachment to host cells, autophagy, invasiveness, biofilm formation, and apoptosis, linking with the host’s immune response. Among other OMPs, Omp33-36 facilitates water movement across the membrane and is linked to host tissue adherence, intracellular invasion, autophagic regulation, and metabolic flexibility in *A. baumannii*. In contrast, OmpW plays a role in colistin interaction and participates in maintaining iron balance within the bacterial cell [251].

Other important outer membrane structures include the carboxylate channel AB1 (OccAB1), which uptakes glycine and ornithine; the OprD protein, which facilitates the diffusion of necessary amino acids into the cell; and CarO, which is crucial for carbapenem resistance [251].

Environmental factors, such as antibiotic treatment, affect OMV protein components. These vesicles frequently contain virulent molecules, including phospholipase C, superoxide dismutase, and catalase—enzymes that facilitate attachment to host cells and promote invasive behavior. Enzymes carried by OMVs bind or absorb antibiotics, aiding in evading drug attacks [251].

LPSs play crucial roles in OMV biogenesis, virulence, and antibiotic resistance. They are also potent activators of immune cells. Alterations in the structure of lipopolysaccharide, particularly the addition of groups like phosphoryl ethanolamine or 4-amino-4-deoxy-L-arabinose to lipid A, reduce the binding affinity of polymyxins and help bacteria withstand their antimicrobial effects. In addition to various types of nucleic acids, including DNA, mRNA, microRNAs (miRNAs) [253] and non-coding RNAs, OMVs are enriched with bioactive compounds that modulate intercellular interactions, shape host defense pathways and support microbial adaptation through nutrient-associated functions [251].

OMVs released by *A. baumannii* have been shown to carry resistance-associated enzymes, including Ambler class C β-lactamases (AmpC) [254] and oxacillinase (OXA) [17], which can enhance bacterial drug resistance through plasmid delivery and other mechanisms. Beta-lactamase, which catalyzes the hydrolysis of β-lactam antibiotics, can be found in OMVs. Notably, OMVs released by *A. baumannii* have been shown to carry zinc-dependent β-lactamases, such as NDM-1, which is selectively secreted by *A*. *baumannii* and exhibits high carbapenemase activity. OMVs contribute to the reduction in β-lactam antibiotic levels in the surrounding environment by delivering enzymes such as class D carbapenemases and AmpC-type β-lactamases, which can inactivate these drugs extracellularly. The level of β-lactamase expression is influenced by bacterial genes. For instance, when the *ISAba1* insertion element is positioned ahead of the *bla_AmpC_* gene, it can upregulate β-lactamase production, thereby increasing bacterial resistance to third-generation cephalosporins [251].

OMVs secreted into the periplasm serve as decoys for membrane-targeting antibiotics. *A*. *baumannii* OMVs can transfer the *bla_OXA-24_* gene, facilitating the spread of carbapenemase genes within *A*. *baumannii*. Additionally, the secretion of NDM-1 into OMVs may help disseminate the *bla_NDM_* gene, illustrating the role of horizontal gene transfer in acquiring extensive drug resistance [251].

Besides transmitting resistance genes, OMVs protect these genes from nuclease degradation. A study using a reconstructed human microbiota model showed that OMVs membrane vesicles derived from *A. baumannii* can serve as protective barriers, reducing the exposure of surrounding bacterial populations to polymyxin B. These results highlight the contribution of OMVs to the development of antimicrobial resistance. Moreover, antibiotics can promote OMVs secretion and modulate OMVs protein components, increasing pathogenicity toward host cells [251].

In addition to mediating antibiotic resistance and host–pathogen interactions, OMVs also influence biofilm development in *A. baumannii*. In vitro experiments have shown that vesicles carrying OmpA and Omp33–36, together with eDNA and enzymes, promote surface adhesion and reinforce biofilm matrix stability [251,252]. Comparative studies further suggest that isolates releasing higher amounts of OMVs tend to display stronger biofilm-forming capacity [251]. Furthermore, OMVs enriched with β-lactamases, including OXA-type and AmpC enzymes, can shield biofilm-embedded bacteria from β-lactam antibiotics by inactivating these drugs extracellularly [17,254]. Taken together, these findings underscore the dual role of OMVs in *A. baumannii*—supporting biofilm persistence while simultaneously spreading multidrug resistance [251,252,254].

### 3.3. OMV Production in P. aeruginosa

The principles of OMV biogenesis in *P. aeruginosa* involve the loss of local connections between the PG layer and the outer membrane, an increase in negatively charged LPS, and elevated periplasmic pressure. The loss of local connections is due to the disruption of covalent bonds between lipoproteins Lpps and PG or changes in protein distribution, resulting in a lack of protein homogeneity and subsequent vesicle formation. This likely occurs because the outer membrane expands more than the PG layer [255].

An increased quantity of negatively charged LPSs arises from higher levels of B-band LPS. The LPS molecule is structurally composed of three main regions: lipid A, a core oligosaccharide segment, and an O-antigen side chain. In *P. aeruginosa*, two distinct O-antigen types are recognized: the A-band, or common polysaccharide antigen (CPA), made up of neutral α-D-rhamnose residues, and the B-band, which represents the strain-specific O-antigen. Accumulation of proteins within the periplasm, particularly in the absence of specific proteolytic enzymes, leads to increased osmotic pressure in this compartment [255]. Notably, only B-band LPS is present in the OMVs of *P. aeruginosa* [241].

This accumulation increases turgor pressure, leading to the generation of OMVs. In conclusion, the formation of OMVs in *P*. *aeruginosa* is complex and involves multiple mechanisms [255].

In *P. aeruginosa*, OMVs are secreted by both free-living and surface-attached cells, primarily in response to PQS-dependent signaling or following cell disintegration and are known to enhance bacterial virulence by participating in pathogenic processes. These vesicles contribute to pathogenicity by preparing host epithelial tissues for colonization, displacing competing microbes at the infection site, and interfering with CFTR (cystic fibrosis transmembrane conductance regulator)-mediated chloride ion secretion in cystic fibrosis airways, which ultimately impairs mucociliary clearance and favors bacterial persistence in the lungs [256].

OMVs derived from *P. aeruginosa* carry a broad array of molecular constituents, including proteins of bacterial origin, various nucleic acids, membrane lipids, and additional biologically active factors. Proteomic analyses of *P. aeruginosa* OMVs have revealed the presence of a broad range of membrane-associated and secreted proteins, including porins (OprD, OprE, OprF, OprG, OprH, OprI), lipid-modifying enzymes (PagL, PcoB), β-lactamases, flagellar components (FlgK, FlgE), folding catalysts such as peptidyl-prolyl cis-trans isomerase, and various outer membrane and transport proteins like LptD, PilQ, EstA, Ggt, OprC, OpdT, FecA, OpdC, OprB, and PslD. Additional factors such as LolB, PonA, OprM, LasA, the lipoprotein NlpD/LppB homolog, PilA, OmlA, WspA, PaAp, and Cif have also been detected [255]. Experimental data have indicated that OprF, a major outer membrane porin in *P. aeruginosa*, contributes to envelope stability by interacting with peptidoglycan and other outer membrane-associated proteins, including OprL and OprI. OprF is suggested to participate in OMV-host cell interactions, potentially contributing to bacterial communication with the host. Additionally, OprD and OprE, which are highly overexpressed in planktonic *P. aeruginosa* and biofilms, are significant. OprD increases bacterial resistance to carbapenems, while OprE contributes to bacterial survival in alkaline environments. Moreover, both OprF and, according to more recent findings, OprH are thought to participate in bacterial attachment and sensing of host-derived signals [257].

Nucleic acids found in OMVs include DNA, RNA, and sRNA. However, the mechanism by which DNA reaches OMVs is not fully understood. Several mechanisms have been proposed to explain how DNA becomes associated with OMVs. One possibility is that extracellular DNA released during bacterial lysis may be taken up through a process resembling natural transformation. Alternatively, DNA might become enclosed within vesicles prior to export via an as-yet unidentified route involving the periplasm. Another explanation involves the presence of outer-inner membrane vesicles (OIMVs), which carry components of both the outer and inner membranes, as well as cytoplasmic material and nucleic acids [255]. sRNAs packed into OMVs can enter host cells and influence the innate immune response [258]. It has been demonstrated that *P. aeruginosa* releases OMVs carrying both sRNAs and virulence-associated molecules, which can be transported into pulmonary epithelial cells [259]. This process alters the methylation patterns of lung macrophages, producing differential CpG methylation at sites associated with cytokines, such as CSF3 [255]. DNA in the OMVs of *P. aeruginosa* promotes horizontal gene transfer, facilitating the dissemination of antibiotic resistance genes among *P. aeruginosa* strains [260].

The virulence arsenal of *P. aeruginosa* encompasses a wide array of factors, including alginate, biofilm-associated components, siderophores such as pyoverdine, LPSs, QS systems, T4P, and various secretion pathways, particularly types II, III, and VI. Among the secreted enzymes are lipases, specific proteases like AprA and PIV, elastolytic enzymes including LasA and LasB, urease, and the cytotoxin exotoxin A. Several of these virulence determinants have also been identified within the cargo of OMVs [256]. LPS within OMVs not only contributes to their virulence [259] but also plays a role in OMV biogenesis and formation [241]. Moreover, LPS molecules can interact with host immune receptors, such as Toll-like receptor 4 (TLR4), initiating inflammatory responses that lead to cytokine production and other immune mediators essential for host defense against infection [261]. Additionally, OMVs transport virulence-associated molecules, including hemolytic phospholipase C, β-lactamase, and alkaline phosphatase, into host cells, further promoting infection and antibiotic resistance [259].

OMVs can fuse with the host plasma membrane via receptor-mediated endocytosis. They can also increase the hydrophobicity of the bacterial cell surface, which in turn promotes biofilm formation. OMVs are regulated by QS systems, enabling bacteria to communicate, colonize, and evade the human immune response [259].

In *P. aeruginosa*, PQS plays a dominant role in quorum sensing by enhancing iron assimilation, chelating ferric ions, and affecting immune-related host responses. Due to its hydrophobic nature, PQS can be incorporated into membrane vesicles, which are actively secreted by bacteria. These vesicles contribute to regulating multiple bacterial processes, including virulence expression, cytotoxic activity, biofilm development, metal ion acquisition, and OMV production [255].

PQS exerts its regulatory role primarily through interaction with PqsR, a LysR-family transcription factor that recognizes the upstream region of the *pqsABCDE* operon. In addition to this receptor-mediated pathway, PQS also engages alternative signaling mechanisms that function independently of PqsR. PQS influences bacterial physiology both transcriptionally and post-translationally: it modulates the expression of multiple genes and can also bind directly to specific target proteins. One example is TseF, an effector of the type VI secretion system (T6SS), whose association with PQS promotes its packaging into OMVs and supports iron transport toward recipient cells. OMVs are critical components of *P*. *aeruginosa* biofilms, with extracellular DNA being a major part of the biofilm matrix [255].

Earlier in vitro studies demonstrated that OMVs are intrinsic components of *P. aeruginosa* biofilms, mediating bacterial co-aggregation and supporting colonization of receptive surfaces [51]. Building on these observations, more recent investigations have shown that PQS-induced OMVs can promote biofilm maturation and, under certain conditions, contribute to biofilm dispersion by delivering proteases and matrix-degrading factors [53,262]. Additional mechanistic evidence indicates that OMVs act as carriers of QS signals such as PQS, reshaping the extracellular matrix and altering its protein/polysaccharide composition, thereby regulating community structure and [53]. Supportive evidence from in vitro coculture models with host epithelial cells supports these roles, while clinical studies have detected OMV-associated material in cystic fibrosis sputum, underscoring their contribution to chronic lung infections. Taken together, these findings highlight OMVs as dynamic modulators of *P. aeruginosa* biofilms, with the capacity to both reinforce and remodel biofilm architecture depending on environmental conditions [51,53,57,262].

During the dispersal stage of biofilm development, the production of both OMVs and PQS reaches its peak. These vesicles carry enzymatic activities, including lipases, nucleases, and proteases, which are thought to assist in matrix breakdown and facilitate the detachment of cells from the biofilm [55]. Exposure to sublethal concentrations of gentamicin has been reported to induce OMV production in *P. aeruginosa*, with some vesicles incorporating the antibiotic. This mechanism may help the bacterial community withstand gentamicin by sequestering it away from its intracellular targets [255,263]. Additionally, OMVs derived from *P. aeruginosa* PAO1 have been shown to provide phage defense, specifically interfering with viral particles like myovirus KT28 and podovirus LUZ7, thereby reducing phage-mediated bacterial killing [255,256].

While OMVs secreted by *P. aeruginosa* can contribute to host tissue damage by transporting resistance-related factors such as β-lactamase, they are also being studied for their biomedical applications, including the delivery of therapeutic agents and immunogens targeting both infectious and neoplastic diseases. Additionally, OMVs play a role in multiple virulence-associated processes, such as facilitating drug resistance, regulating bacterial density, and evading the host immune response [259].

### 3.4. OMV Production in K. pneumoniae

Like many Gram-negative bacteria, *K*. *pneumoniae* produces and secretes OMVs into the extracellular environment [264]. OMV formation during bacterial growth and division may be driven by localized weakening of structural contacts either between the outer membrane and PG layer or involving a broader detachment between the outer membrane, PG, and inner membrane, ultimately leading to membrane protrusion and vesicle release. However, these mechanisms do not explain how elements of the inner membrane appear in OMVs. It has been established that *K. pneumoniae* produces OMVs ranging from 20 to 200 nm in diameter. Proteomic analyses have identified a variety of proteins within *K. pneumoniae*-derived OMVs, such as OmpX, lipoproteins associated with cell wall integrity, stress-related phage shock proteins capable of inducing protective responses, uncharacterized proteins like YgdR, and ribosomal subunit components including the 30S protein S20. These identified proteins are among the most commonly detected components originating from different cellular locations, such as the outer and inner membranes, the periplasm, the extracellular milieu, and the cytoplasmic compartment. This indicates that *K. pneumoniae* OMVs contain numerous proteins originating from various cellular compartments [265].

The key components of OMVs in *K. pneumoniae* include OMPs and porins, such as OmpA [265] and OmpC (also referred to as OmpK36) [265,266]. OmpA is an essential membrane protein that contributes to bacterial pathogenicity by facilitating adhesion and invasion, while also maintaining the structural integrity of the bacterial envelope through anchoring to the peptidoglycan layer. Furthermore, OmpA participates in the transfer of molecules. Notably, the absence of OmpA in *K. pneumoniae* results in increased susceptibility to polymyxin [267,268]. A detailed analysis of OMVs produced by the *K. pneumoniae* strain KpHCD1, isolated from a urine culture, has revealed that OmpA serves as a virulence factor by protecting the bacterium against the innate immune response [40]. Overexpression of OmpC (OmpK36) enhances bacterial resistance, particularly to β-lactam antibiotics, by reducing outer membrane permeability [269]. Additionally, porins play a role in the immune response. *K*. *pneumoniae* mutants that lack both OmpK35 and OmpC (OmpK36) exhibit altered OMV-associated protein compositions when compared with strains that retain expression of one or both porins. Absence of OmpC (OmpK36), when accompanied by β-lactamase production, has been linked to elevated levels of carbapenem resistance, especially among strains producing extended-spectrum β-lactamases (ESBLs). Loss of porins also alters macrophage inflammatory responses to OMVs, resulting in significantly reduced pro-inflammatory cytokine secretion [264,270]. Another important protein contributing to virulence and AMR is OmpK37 [270].

Experimental studies using human bronchial epithelial BEAS-2B cells have demonstrated that *K. pneumoniae* OMVs provoke a robust inflammatory reaction, characterized by upregulated transcription of pro-inflammatory cytokines like IL-6, IL-8, IL-1β, and TNF-α, along with observable shifts in miRNA expression patterns. In vivo, inoculation with OMVs from hypervirulent *K. pneumoniae* resulted in increased levels of the pro-inflammatory chemokines IL-6, IL-8, and TNF-α [264]. Among virulence factors associated with *K. pneumoniae* strain HCD1 (KpHCD1) OMVs, type 1 and type 3 fimbriae genes have been identified, which are important for *K. pneumoniae* adhesion and biofilm formation. Several genes for enterobactin, the primary iron uptake system in *K. pneumoniae* infection, have also been identified in the KpHCD1 genome. Enterobactin contributes to virulence. Additionally, three β-lactamases—two ESBLs such as Temoneira-1 (TEM1) [271] and cefotaxime Munich-15 (CTX-M-15) β-lactamases [272] and *K. pneumoniae* carbapenemase (KPC-2) [17]—have been found in KpHCD1 OMVs, suggesting that carriage of plasmids harboring ESBL genes may contribute to enhanced virulence potential of the strain [270]. *K. pneumoniae* can deliver plasmid-encoded AMR and virulence genes to other species, such as *E. coli*, via horizontal gene transfer [273].

Efflux transport systems such as AcrAB and OqxAB, known for expelling a broad range of antibiotics and enhancing bacterial aggressiveness, were also identified. Exposure to meropenem and polymyxin has been linked to increased expression of the AcrB subunit, a component of the multidrug efflux system, which coincides with heightened drug resistance and virulence in *K. pneumoniae*. Efflux systems embedded in OMVs might contribute to trapping antibiotics outside bacterial cells, thereby reducing their effective concentration in the surrounding environment [270].

Other significant proteins found in KpHCD1 OMVs are two key elements involved in the Lpt-mediated LPS transport pathway, namely LptD and MsbA. LptD was consistently observed under all tested antibiotic treatments, while MsbA was more specifically detected following exposure to meropenem, polymyxin, or their combination with amikacin—agents that disrupt bacterial envelope architecture [270].

LptD and LptE form a functional complex that directs LPS to the outer membrane and supports its proper assembly at the bacterial surface, while MsbA facilitates the relocation of lipid A to the outer leaflet of the outer membrane. Their enrichment in OMVs may reflect elevated expression and membrane accumulation triggered by antibiotics that interfere with cell envelope integrity. Another LPS modification-related protein, ArnT, was found in OMVs under antibiotic exposure. ArnT is a transferase that adds an L-Ara-N molecule to the lipid A portion of LPS, conferring resistance to polymyxin [270].

Many ribosomal proteins have been identified in KpHCD1 OMVs: 21 from the 30S subunit and 15 from the 50S subunit. Treatment with polymyxin B appears to modulate gene expression and protein production, as suggested by the detection of RNA polymerase sigma factor RpoD and the observed upregulation of its alpha and beta subunits within OMVs following antibiotic exposure. Enhanced vesiculation observed in KpHCD1 during meropenem exposure, together with the increased presence of envelope-modifying proteins like NlpD, YbiS, MltA, LpoA, and TolB, suggests that this β-lactam antibiotic may interfere with membrane remodeling pathways involved in OMV generation [270].

It has also been observed that treatment with polymyxin B in both polymyxin-susceptible and extremely drug-resistant *K*. *pneumoniae* strains results in the shedding of OMVs containing proteins involved in OM remodeling, such as those related to LPS biosynthesis, and factors associated with the bacterium’s pathogenic potential, including QS. Interestingly, expression levels of β-lactamases such as sulfhydryl variable lactamase (SHV) [17] and TEM markedly declined in both strains following treatment, implying that exposure to polymyxin B could restore bacterial susceptibility to β-lactam antibiotics by altering resistance mechanisms [267].

Thus, OMVs are involved in pathogenicity, survival, stress response, and resistance dissemination. *K*. *pneumoniae* employs OMVs to defend against antibiotics and facilitate cooperative interactions within microbial populations, reinforcing the concept that cargo loading into OMVs follows a regulated, non-random pattern [270].

In *K. pneumoniae*, OMVs are increasingly recognized as modulators of traits that support biofilm-associated lifestyles. OMVs delivered during growth carry protein and LPS cargo that can activate host innate responses and remodel the infection microenvironment, processes that may indirectly favor colonization and persistence [264,265]. Under antibiotic stress, OMV cargo composition shifts, indicating regulated loading that could influence survival of surface-attached communities [270]. Importantly, OMVs act as vehicles for horizontal gene transfer: *bla*NDM-1 has been shown to be packaged and transferred via OMVs between *K. pneumoniae* strains, a mechanism that can bolster the resilience of biofilm-embedded populations under antimicrobial pressure [274,275]. More broadly, a recent review of bacterial extracellular vesicles outlines how vesicles can participate across the biofilm life cycle—from initial attachment and matrix modulation to dispersion—suggesting analogous roles for *K*. *pneumoniae*, even as direct, species-specific adhesion/matrix-stabilization assays remain comparatively limited [56].

### 3.5. OMV Production in Enterobacter spp.

*E*. *cloacae* can produce OMVs via outer membrane blebbing. Additionally, some vesicles possess a double-membrane-containing structure, indicating that *E*. *cloacae* also secrete OIMVs. The sizes of *E*. *cloacae* OMVs range from 20 to 300 nm, with an average diameter of approximately 150 nm. The protein cargo of *E. cloacae* OMVs is functionally diverse and includes transport-associated proteins, membrane receptors, signaling elements, metabolic enzymes, stress-adaptive proteins, and components related to glucose catabolism. In addition, proteins involved in PG biosynthesis, metabolic regulation, and host–bacterium interaction are also frequently identified. In total, 229 proteins have been identified in *E*. *cloacae* OMVs [276].

Among the most abundant proteins are flagellin, putative outer membrane porin, putative outer membrane lipoprotein, type I fimbrial protein subunit A, Gp12, Tal-Pal system TolB protein, major tail sheath, phage major capsid, and DNA binding transcriptional activator OsmE (osmotically inducible lipoprotein OsmE) [276].

Notably, OmpX ranks as the 12th most frequently detected protein within *E. cloacae* OMVs and is regarded as a potential contributor to pathogenicity. *E*. *cloacae* cells interact in a sessile manner (rather than planktonic) even at diluted concentrations, leading to biofilm formation. The presence of OmpX in OMVs enhances biofilm formation. In addition to its role in biofilm formation, OmpX contributes to virulence, suggesting that OMVs may carry pathogenic determinants. Other host-interaction-related proteins include OmpA family lipoprotein, putative metalloprotease YggG, BOF domain-containing protein, and putative immunoglobulin domain protein [276].

OMVs are enriched with hydrolase-type enzymes that mediate the breakdown of complex substrates into simpler molecules by incorporating water during bond cleavage. Within bacterial cells, hydrolases contribute to various physiological processes, including the remodeling of PG during cell elongation, division, and programmed cell lysis. From the host side, hydrolases released by bacteria may support communication between microbial and epithelial cells, modulate local immune responsiveness, and alter epithelial barriers by promoting the translocation of bacterial products across intestinal tight junctions [276].

Proteins responsible for PG synthesis and metabolism include endolytic peptidoglycan transglycosylase RlpA, penicillin-binding protein (PBP) activator LpoA, Lpp repeat-containing protein (major outer membrane lipoprotein), putative lipoprotein, and putative peptidase M23B family protein. Stress response-related proteins found in OMVs include copper lipoprotein, uncharacterized protein, periplasmic protein, periplasmic trehalase, and penicillin-binding protein-1b (PBP-1b) [276].

Additionally, fourteen proteins involved in bacteriophage structural formation, including those contributing to head organization, contractile tail components and host cell rupture, have been detected, along with two proteins likely functioning as attachment points for phage particles. These proteins likely integrate into the *E. cloacae* genome and are consistently produced throughout bacterial growth, allowing their inclusion in OMVs [276].

*E*. *cloacae* OMVs contain periplasmic, extracellular-membrane, and cytoplasmic proteins, including 11 ribosomal proteins that assist in protein translation. These findings suggest that explosive cell lysis may also be involved in OMV biogenesis. Genomic analysis of OMVs corresponds to the biological strain of *E*. *cloacae*. OMVs can accumulate on surfaces to form a matrix-like substance and structure, indicating their role in matrix-assisted biofilm formation [276].

Based on the published findings, OMVs in *Enterobacter* spp. are not only stress-adaptive structures but also active modulators of biofilm-associated behavior. OmpX-enriched vesicles enhance initial adhesion, while OMV accumulation on abiotic surfaces promotes the formation of matrix-like structures that stabilize biofilms. Their diverse enzymatic and signaling cargo suggests roles in remodeling the extracellular matrix and mediating host–pathogen interactions, thereby strengthening the persistence and resilience of *Enterobacter* biofilms [276].

The main components of OMVs in Gram-negative members of the ESKAPE group are listed in Table 3.

## 4. Concluding Remarks

Biofilm and OMV formation in Gram-negative ESKAPE pathogens present a major obstacle in the fight against AMR. Far from being simple defensive maneuvers, these processes actively contribute to the survival and increased virulence of these bacteria in environments that include exposure to antibiotics. Biofilms serve as protective shields, safeguarding bacterial communities from both the immune system and antimicrobial agents. At the same time, OMVs contribute to bacterial defense and aggression by carrying enzymes that break down antibiotics and delivering virulence factors to host cells.

Biofilm formation follows several distinct phases: initial attachment, microcolony formation, maturation, and dispersion. While these phases share similarities across bacterial species, they can involve different signaling molecules and regulatory proteins, such as various autoinducers used in QS, depending on the specific bacterial strain. For instance, Gram-negative bacteria like *P. aeruginosa* rely on the signaling molecule PQS, while *K. pneumoniae* may use other QS signals. These differences in signaling molecules help regulate the biofilm’s structure and resistance properties.

Similarly, the formation of OMVs follows a series of stages, beginning with the production of membrane blebs, followed by detachment, and, in some cases, the fusion with the host membrane. Just as in biofilm formation, proteins and molecules involved in OMV production differ between bacterial species. For example, the presence of specific enzymes such as β-lactamases in OMVs may vary, which enhances the bacterium’s ability to resist antibiotics and propagate resistance mechanisms.

Importantly, current evidence shows that the interplay between OMVs and biofilms is bidirectional. OMVs promote adhesion, matrix stabilization, and dispersal, while the biofilm environment itself modulates vesicle production and cargo loading. This reciprocal influence reinforces bacterial adaptation and resilience under stress.

At present, most experimental data linking OMVs and biofilm development come from studies on *A. baumannii* and *P. aeruginosa*, while much less is known about *K. pneumoniae*, *Enterobacter* spp., and other Gram-negative ESKAPE pathogens. Future research should therefore expand to these underexplored bacteria to clarify whether similar or distinct OMV–biofilm interactions occur across species.

The complex functions of biofilms and OMVs underscore the resilience and adaptability of Gram-negative bacteria, which makes them particularly challenging to combat in clinical environments. Understanding the molecular mechanisms behind biofilm and OMV production is essential to developing new treatments that can effectively target these processes. Disrupting biofilm formation and OMV production has the potential to increase the effectiveness of current antibiotics and slow the advance of resistance.

Moreover, the exploration of OMVs in the development of vaccines and drug delivery systems offers a promising avenue for innovative treatments that could leverage these bacterial strategies to our advantage. In summary, overcoming the challenges associated with biofilm and OMV formation in Gram-negative ESKAPE pathogens requires a comprehensive approach that integrates advanced research with novel therapeutic innovations. By focusing on these critical aspects of bacterial pathogenesis, meaningful advances may be achieved in tackling the growing threat of AMR and enhancing clinical success in the management of these pathogens.

## Figures and Tables

**Figure 1 ijms-26-09857-f001:**
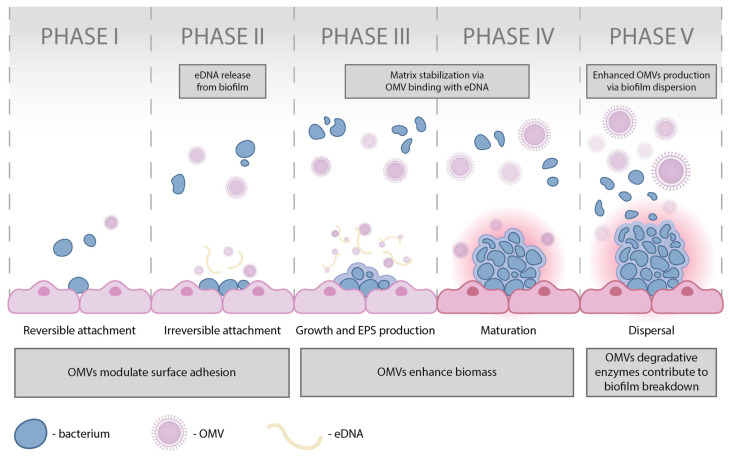
Five biofilm development phases and OMV roles (adapted from source [38], distributed under the condition of the Creative Commons CC BY license). OMV—outer membrane vesicle, eDNA—extracellular DNA.

**Figure 2 ijms-26-09857-f002:**
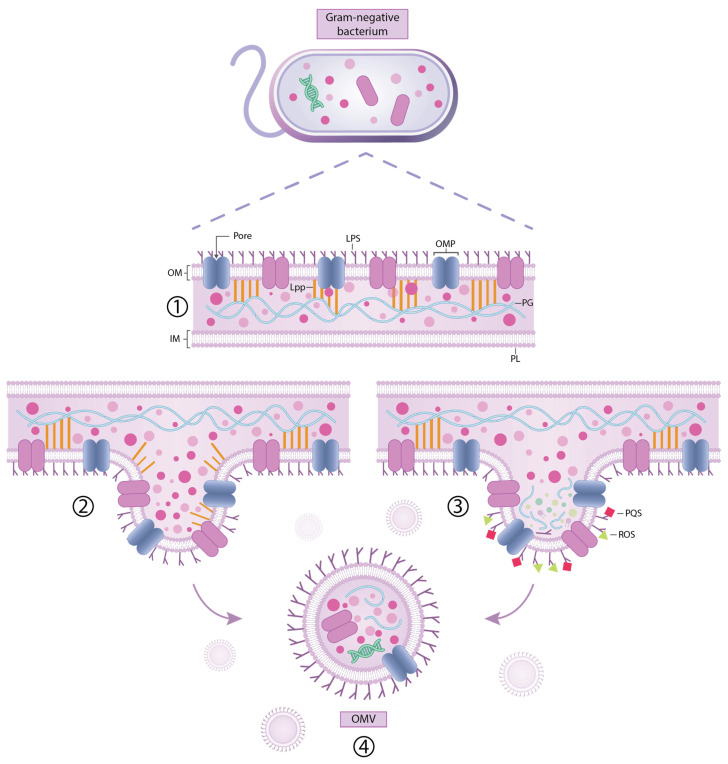
Formation of OMVs. 1. Normal state: the OM is anchored to the PG layer via Lpp maintaining envelope stability. 2. Pathway I: reduction or loss of PG–Lpp crosslinking decreases OM stabilization, resulting in local bulging and vesicle release. 3. Pathway II: periplasmic accumulation of misfolded proteins, PG fragments, LPS, or stress-induced molecules such as PQS and ROS generates local pressure on the OM, driving vesiculation. 4. Released OMV: the final vesicle contains outer membrane components (lipids, proteins, LPSs, LOSs), periplasmic material, and various virulence factors. LOS—lipooligosaccharide; Lpp—lipoprotein that staples OM to PG to maintain the structural integrity of the cell envelope; LPS—lipopolysaccharide; OM—outer membrane; OMP—outer membrane protein; OMV—outer membrane vesicle; PG—peptidoglycan layer; PL—phospholipid; PQS—Pseudomonas Quinolone Signal; ROS—reactive oxygen species (adapted from source [38], distributed under the condition of the Creative Commons CC BY license).

**Figure 3 ijms-26-09857-f003:**
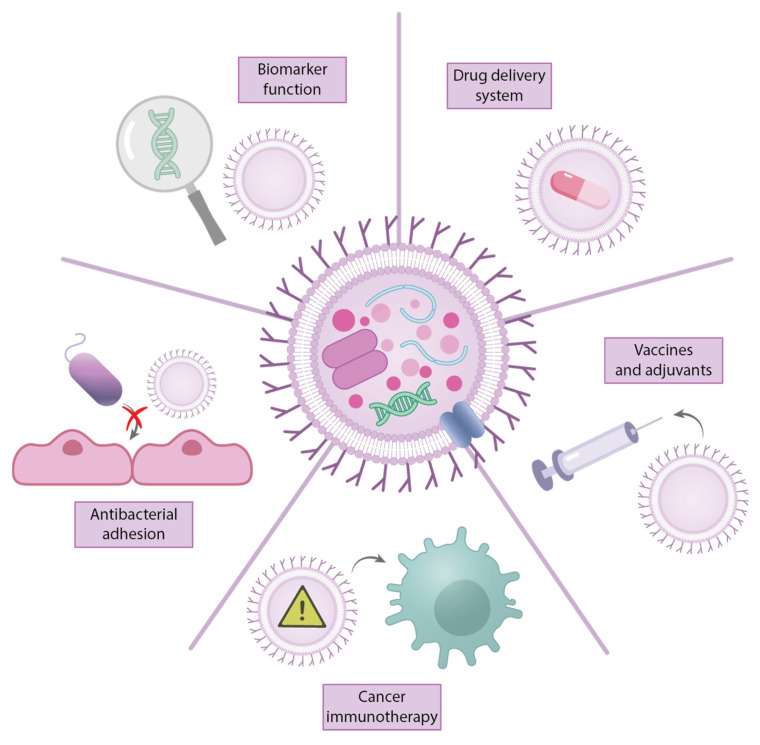
Main applications of OMVs.

**Table 1 ijms-26-09857-t001:** AMR trends in Gram-negative ESKAPE pathogens in the context of the COVID-19 pandemic across the countries reported (adapted from source [13], distributed under the condition of the Creative Commons CC BY license), with additional data on *E. cloacae* in Saudi Arabia from source [22]. AMG—aminoglycosides, FQs—fluoroquinolones, IQR—interquartile range.

	Antibiotic	*A. baumannii* Median % (IQR)	*K. pneumonia* Median % (IQR)	*P. aeruginosa* Median % (IQR)	*E. cloacae* (N = 638) Resistant %
Beta–lactam antibiotics	Amoxicillin/ clavulanate	–	81 (79.3–83.75)	–	99.84
	Ampicillin	–	100 (90.5–100)	–	100
	Aztreonam	–	84.7 (67.27–88.87)	–	–
	Cefazolin	–	93 (78–95.5)	–	–
	Cefuroxime	–	88.9 (79.6–91.42)	–	79.31
	Cefepime	94.4 (93–100)	81.15 (71.7–87.25)	14.3 (12.5–47.8)	13
	Ceftazidime	91.2 (50–100)	93.5 (83.7–97.9)	40 (23–41.7)	23.2
	Cefoperazone/ sulbactam	–	76.2 (73.8–77.9)	–	–
	Ceftriaxone	76.2 (54.75–95.55)	84 (77.55–93.4)	75 (43.75–87.5)	34
	Meropenem	92.1 (64.02–95.65)	71.25 (55.37–77.37)	38 (18.37–42.17)	9.56
	Imipenem	92.1 (80.65–95.72)	65.7 (19.25–72.87)	42.9 (19.75–52.9)	9.87
	Ertapenem	–	71.4 (55.55–75.05)	–	9.25
	Piperacillin/ tazobactam	93.7 (66.9–100)	77.7 (57.1–79.27)	11.25 (9.25–13.85)	13.64
AMG	Amikacin	84.6 (56.3–92.95)	69.85 (58.7–80.12)	25 (12–28)	–
	Gentamicin	95.7 (74.2–97.1)	57.1 (33.45–86.6)	25 (19.75–58.75)	10.7
FQs	Levofloxacin	97.05 (91.92–100)	80.8 (78.55–90.85)	43.5 (28.6–80)	7.4
	Ciprofloxacin	91.2 (65–100)	87.8 (55.1–92.95)	50 (32.3–62.5)	10.5
Other antibiotics	Trimethoprim/ sulfamethoxazole	50 (46.8–84.2)	73.5 (32–74)	–	22.8
	Tigecycline	9.5 (8.8–33.3)	31.4 (1.7–44)	–	–
	Nitrofurantoin	–	51.8 (38.5–60.6)	–	–
	Colistin	2.5 (0–19.62)	21.1 (12.42–69.82)	4 (0–12.25)	–

**Table 2 ijms-26-09857-t002:** Core regulatory components and structural contributors in Gram-negative ESKAPE biofilms.

Bacterium	Biofilm Forming Factors	Main Function
*A. baumannii*	Fimbriae [64,72,73]	Bacterial mobility [72,79,82,83,84]
	Pili (type I and type IV) [72]	
	Surface-adhesion proteins Baps [64,72,73]	Biofilm initiation and maturation [73,104,105]
	Csu pilus (CsuA/BABCDE-mediated pilus) [64,67,72,86,87]	Recognition of adhesion site and biofilm formation on the abiotic surfaces [64,72,91]
	TCS BfmRS [72,92]	Regulation of Csu pili operon activity, with downstream impact on cellular structure, surface-associated biofilm architecture, adhesion to inert and biological substrates, and antimicrobial tolerance [64,72,93,94]
	Autoinducer synthase AbaI [73,95,97,98,99,100]	Production of signaling molecules involved in QS, supporting normal biofilm development and influencing the expression of antimicrobial resistance determinants, efflux transport components, and motility-associated functions [73,95,97,98,101]
	QS, including autoinducer synthase AbaI, receptor protein activator AbaR, along with N-acyl-homoserine lactones (AHLs) [73,99,100]	Cell-to-cell communication [101]
	*blaPER–1* gene [73]	Contribution to *A. baumannii* adhesion to respiratory tract linings and development of surface-associated communities on host-derived and inert materials, promoting bacterial persistence and pathogenic potential [73,100]
	*epsA* locus, associated with outer membrane-bound EpsA protein implicated in exopolysaccharide matrix formation; *ptk* locus, coding for a predicted tyrosine kinase involved in signal transduction and exopolysaccharide regulation [73,100,108,109]	Enhancing the bacterium’s ability to form robust biofilms [73]
	Outer membrane porin OmpA [73]	Adhesion to epithelial surfaces, intracellular access mechanisms targeting mitochondrial integrity, and promotion of host cell demise [73,111,112,113,114] Contribution to biofilm formation on abiotic and biotic surfaces [73]
	*bfmS* gene, encoding the cytoplasmic membrane sensor kinase BfmS as part of the TCS BfmRS [73]	Mutations in this gene cause disruption of disrupt AbOmpA regulatory pathways [73]
	eDNA [72,116]	Contribution to EPS layer assembly [72,117,118,119]
	PNAG, encoded by the *pgaABCD* gene cluster [72,120,121]	Facilitation of biofilm structuring, bacterial attachment, surface-to-cell adhesion, cell-to-cell interactions, pellicle formation, and contributes to bacterial protection against innate host defenses [72,120,121]
	c-di-GMP [72,122]	Regulation of biofilm formation and surface-associated motility) [72,122]
*P. aeruginosa*	Flagellum [131,132]	Initiation of biofilm formation [131,132]
	Type IV pili [130,134]	Mediation of irreversible cell-to-surface colonization [130,134]
	c-di-GMP, regulated by the Wsp chemosensory-like signal transduction pathway, and promoted by the proteins WspR are FleQ and PelD [130,135]	Production of matrix components, including exopolysaccharides [130,131]
	Exopolysaccharides such as alginate, Pel, and Psl [130,131,136,139,140,141]	Contribution to surface attachment and the stability of biofilm architecture [130,131,136,139,140,141]
	eDNA [130,136]	Nutrition source for the bacteria within biofilm, cellular organization support, cation chelation, establishment of an optimal environment for bacteria residing in biofilms [131,136,160,161,162,163]
	Extracellular adhesin CdrA, secreted with the CdrA- CdrB TCS [130,164]	Bacterial adhesion and preservation of biofilm structural integrity [130,165]
	PHA [136,166]	Implication in stress tolerance and attachment to abiotic surfaces like glass, possible contribution to energy-generating metabolic processes [136,166,167,168]
	QS circuitry involving Las, Rhl, and PQS-MvfR signaling modules with their respective regulatory elements [131,169,170,171]	Development of mature and differentiated biofilms, also regulation of various processes that contribute to severe systemic infections [131,172,173]
	TCSs GacS/GacA and RetS/LadS [173,174]	Contribution of TCS GacS/GacA to biofilm formation, bacterial fitness, motility, stress tolerance, and virulence [173,175,176,177] Contribution of RetS to the repression of biofilm formation, and mediation of biofilm development by LadS [179,180,181,182,183,184,185]
	Cyclic glycerophosphorylated β-(1,3)-glucans [131,186]	Contribution to AMR [131,187,188,189]
*K. pneumoniae*	Type III fimbriae (more dominant) [190]	Biofilm formation: contribution to bacterial adhesion on non-biological surfaces, and modulation of the c-di-GMP-governed shift in growth phenotype from motile to sessile state [190,202,203] Regulation of type III fimbrial expression supporting community establishment, associated with *mrkA*, *mrkD*, and the *mrkHIJ* genetic locus [190,197,198,199,200,203]
	Type I fimbriae [190]	Biofilm formation, mainly regulated by *fimK*, *fimH*, *kpfR* genes [36,190,195]
	*E. coli* common pilus (ECP) fimbriae [36,190]	Participation in cell adhesion, biofilm formation, and colonization of various environments [190]
	Polysaccharide capsule [190,205,206]	Contribution to multiple stages of biofilm formation, including initial adhesion and maturation [190,205,206] The main genes responsible for polysaccharide synthesis include *wza* homologous, ORF14, *treC*, and *sugE* [36,205,207,208]
	LPSs [36,208]	Contribution to the initial attachment of *K. pneumoniae* to abiotic surfaces [36,208]; The main genes responsible for this process are *wzm* and *wbbM* [36,208]
	Virulence gene *wcaG* [190,209]	Involvement in capsule biosynthesis and biofilm formation [190,209]
	Type II QS pathway based on AI-2 signaling molecules synthesized via LuxS-mediated catalytic activity [36,190,205,212,213]	Regulation of fimbriae, exopolysaccharide, adhesin, and other substance synthesis via signaling molecules, promoting biofilm formation and maturation in bacteria [190,210,211]; Facilitation of interspecific communication, enabling bacteria to respond to both autoinducers: AI-2 produced by other species and their own AI-2 [36,190,205,212,213]; The primary gene responsible for type II QS activity is *luxS* [36,212]
	Efflux pumps AcrA, OqxA, QacEΔ1, and EmrB [205,216]	Contribution to biofilm formation: enhancing antibiotic resistance, regulating exopolysaccharide production, and promoting bacterial survival within the biofilm matrix [205,216]; The primary genes responsible for efflux pump function are *acrA*, *emrB*, *oqxA*, and *qacEΔ1* [205,218]
	Fur [36,195]	Iron balance regulation within biofilm-associated populations through modulation of genetic pathways linked to iron acquisition and metabolic processing, with downstream effects on community structure, stability, and bacterial virulence [36,195,202,221]
*Enterobacter* spp.	Curli fimbriae [223,226]	Attachment of bacterial cells to non-living materials and host-associated surfaces [223,226]
	Secondary type VI secretion apparatus (T6SS-2) [223,227]	Participation in surface association with inert materials and host-derived substrates [223,227]
	*csgA* and *cdgD* genes [223,228]	Regulation of biofilm formation and control through curli biogenesis [223]
	Autoinducers AI-1, AI-2, and AI-3 [223,229]	Decreased bacterial adhesion and biofilm downregulation due to AI-1 [229,230]; Intercellular communication in *Enterobacter* spp. via Lsr-type receptors induced by AI-2 [223,231,232]; Facilitation of interspecies QS and regulation of biofilm formation through Lsr-type receptors [229,233]

**Table 3 ijms-26-09857-t003:** Key components of OMVs in ESKAPE Gram-negative bacteria.

Bacterium	OMV Components	Functions
*A. baumannii*	Phospholipids and lipopolysaccharides [250]	Maintenance morphology and stability of outer membrane vesicles [250]
	Outer membrane protein OmpA [251,252]	Determination of vesicle formation sites in the outer membrane, promotion of OMV-mediated pathogenesis, and induction of host cell death, acting as a major virulence factor in humans [251]
	Outer membrane protein Omp33-36 [251]	Adhesion, invasion, regulation of autophagy, metabolic adaptability [251]
	Outer membrane protein OmpW [251]	Colistin binding, regulation of bacterial iron homeostasis [251]
	Outer membrane structure OccAB1 (Carboxylate channel) [251]	Uptaking glycine and ornithine [251]
	Outer membrane protein OprD [251]	Facilitating the diffusion of necessary amino acids into the cell [251]
	Outer membrane protein CarO [251]	Carbapenem resistance [251]
	Phospholipase C, superoxide dismutase, and catalase [251]	Promotion of surface binding and invasive potential [251]
	Nucleic acids (DNA, mRNA, miRNA, non-coding RNA) and signaling molecules [253]	Cellular communication, host immune responses, and bacterial nutrition [253]
	Enzymatic resistance factors: Ambler class C β-lactamases (AmpC), oxacillinase (OXA), and New Delhi metallo-β-lactamase-1 (NDM-1) [251]	Antimicrobial resistance [251]
	*bla_OXA-24_* and *bla_NDM_* genes [251]	Facilitating the dissemination of antimicrobial resistance through OMVs [251]
*P. aeruginosa*	Lipopolysaccharides [255]	Immune system activation, contribution to virulence, participation in the OMV biogenesis and formation [255]
	Outer membrane protein OprF [255,257]	Possible contribution to OMV interaction with host cells and recognition of external signals from host cell [255,257]
	Outer membrane protein OprD [255,257]	Bacterial resistance to carbapenems [255,257]
	Outer membrane protein OprE [255,257]	Contribution to bacterial survival in alkaline environment [255,257]
	Outer membrane protein OprH [255,257]	Adhesion to the host cell and recognition of external signals from the host cell [255,257]
	Small RNA [255]	Entering host cells and influencing the innate immune response [258]
	DNA [260]	Promotion of horizontal gene transfer, facilitating the dissemination of antibiotic resistance genes among *P. aeruginosa* strains [260]
	Lipase, nuclease, and protease [256]	Contribution to the extracellular degradation of matrix components during biofilm dispersion [256]
	Pseudomonas quinolone signal (PQS) [255]	Contribution to the modulation of virulence traits, biofilm development, iron acquisition, cytotoxicity, and OMV production [255]
*K. pneumoniae*	OmpA protein embedded in the outer membrane [265]	Immune system activation, contributing to virulence by protecting the bacterium from innate immune responses [267,268]
	OmpC (OmpK36), a porin embedded in the bacterial outer membrane [265,266]	Enhancement of resistance to β-lactam antibiotics by reducing outer membrane permeability, contribution to carbapenem resistance, particularly in ESBL-positive strains, alteration of the immune response by affecting macrophage inflammatory responses to OMVs, resulting in reduced cytokine secretion [264,269,270]
	Outer membrane protein OmpK37 [270]	Contribution to virulence and antimicrobial resistance [270]
	Type 1 and type 3 fimbriae genes [271]	Contribution to virulence, bacterial adhesion and biofilm formation [271]
	Enterobactin [247,271]	Contribution to virulence [271]
	TEM1, CTX-M-15 and KPC-2 [17,271,272]	Contribution to virulence [270]
	Efflux pumps AcrAB and OqxAB [270]	Mediation of antimicrobial resistance and contribution to virulence [270]
	LptD, MsRA, LptE, ArnT (proteins from lipopolysaccharide transport machinery) [270]	Contribution to maintaining outer membrane integrity and resistance to antimicrobial agents, including polymyxins [270]
	Ribosomal proteins (21 from 30S, 15 from 50S), RNA polymerase sigma factor RpoD, RNA polymerase alpha and beta subunits, NlpD, YbiS, MltA, LpoA, TolB proteins [270]	Contribution to resistance to polymyxin B through transcription and translation disruption, participation in vesiculation, which is enhanced by meropenem exposure [270]
	QS factors [17]	Involvement in pathogenic potential [17]
*Enterobacter* spp.	Outer membrane protein OmpX [276]	Contribution to virulence and biofilm formation [276]
	OmpA family lipoprotein, predicted metalloprotease YggG, BOF-like domain protein, and immunoglobulin-like domain protein [276]	Involvement in bacterial–host engagement [276]
	Hydrolases [276]	Facilitation of cell interaction, enhancement of immune tolerance, and mediation of microbial compound transfer across the intestinal epithelial barrier [276]
	Endolytic peptidoglycan transglucosylase RlpA, PBP activator LpoA, Lpp-repeat motif-containing lipoprotein, predicted outer membrane lipoprotein, M23B family peptidase [276]	Involvement in peptidoglycan synthesis and metabolism [276]
	Copper lipoprotein, uncharacterized protein, periplasmic protein, periplasmic trehalase, and penicillin-binding protein-1b (PBP-1b) [276]	Involvement in stress response [276]

## Data Availability

No new data were created or analyzed in this study. Data sharing is not applicable to this article.

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
