# Peer review of "Biofilm and Outer Membrane Vesicle Formation in ESKAPE Gram-Negative Bacteria: A Comprehensive Review"

_ijms, 2025, doi:10.3390/ijms26209857_

Round 1

Reviewer 1 Report

Comments and Suggestions for Authors

In presented paper, the Authors discuss in detail the mechanisms of biofilm and OMV formation and their significance, using ESKAPE Negative Bacteria as an example. Beyond the significance of this group of microorganisms, which is responsible  for approximately 80% of hospital infections, Authors discuss the problem of bacterial drug resistance, which is largely determined by their ability to form biofilm and produce OMVs. The work is well organized, which allows for a clear and readable presentation of all aspects (including molecular ones) related to biofilm and OMV formation. The advantage of the presented manuscript is the discussion of the, as yet poorly understood, process of OMV biogenesis and the OMVs functions , including protection against antibiotics. Therefore, I recommend the publication of the  manuscript in present form.

Author Response

Dear Rewiever 1

We would like to express our sincere gratitude for your positive evaluation of our manuscript. We are very pleased that you considered our work valuable, and we truly appreciate your encouraging feedback.

With kind regards,

Authors

Reviewer 2 Report

Comments and Suggestions for Authors

This review offers a clear and informative overview of the formation of biofilm and OMV in ESKAPE-Negative Bacteria, Klebsiella pneumoniae, Acetinobacter baumannii, Pseudomona aeruginosa, and Enterobacter spp and their contribution to antimicrobial resistance (AMR).Biofilms act as protective barriers, shielding bacterial communities from both host immune responses and antimicrobial agents. Concurrently, OMVs play dual roles in bacterial defense and offense by transporting enzymes that degrade antibiotics and delivering virulence factors directly to host cells.

Fig1 should be a table not Fig, and it didn’t contain the Enterobacter spp.

Table 1, summary of key factors involved in biofilm formation, it would be better to include the references to the corresponding factors and mark the specific phase in the biofilm formation process to which these factors belong.

Table 2, key components of OMVs in ESKAPE, it would be better to include the references to the corresponding components.

Fig 3 should be reused from reference 31. And OMVs has two mechanism, Fig 3 seems don’t contain all this information.

Line 64 „syndemic“ should be “syndemic”

Reference 14 and 16 are the same.

Author Response

Dear Reviewer 2

We sincerely thank you for your careful review and constructive suggestions, which have been very valuable in improving our manuscript. Following your comments:

  1. The error in referencing has been corrected.
  2. The section on ESKAPE-negative Enterobacter spp. has been supplemented with the available information on E. cloacae. The details are provided both in the main text and in Table 1.
  3. As suggested, Figure 1 has been renamed as Table 1.
  4. References have been listed in Tables 2 and 3, and both tables were reorganized to follow the logical sequence of the manuscript text.
  5. Figure 3 has been revised with more precise and detailed explanations to better illustrate the mechanism of OMV formation.

We highly appreciate your valuable insights, which helped us to significantly improve the clarity and quality of our work.

With kind regards,

Authors

Reviewer 3 Report

Comments and Suggestions for Authors

Dear authors

Thank you for the opportunity to review your manuscript. I appreciate the effort and expertise that went into this work.

General comments

  • The manuscript is a well-written and comprehensive review that effectively synthesizes current knowledge of molecular mechanisms underlying biofilm formation in ESKAPE Gram-negative pathogens and OMVs formation. The content is dense with technical detail, appropriate for a specialized scientific audience, though it may be less accessible to general readers. However, the manuscript lacks mechanistic depth in several key sections. For instance, in lines 541–585, while gene expression changes are noted, the downstream effects on biofilm matrix composition or host-pathogen interactions are not thoroughly explored. Similarly, in lines 283–289, the authors report that biofilm-forming strains are more susceptible but do not provide a mechanistic explanation for this observation.
  • A recurring issue throughout the manuscript is the absence of experimental context. Many findings are presented without specifying whether they are derived from in vitro experiments, in vivo studies, or clinical isolates. This omission diminishes the translational relevance of the review. For example, in lines 283–289, there is no information about the origin of the clinical strains or the types of infections involved.
  • While the manuscript defines its scope well, it falls short of offering a critical or analytical perspective on the role of biofilm formation and OMVs in ESKAPE pathogens. The authors primarily summarize existing literature without introducing novel interpretations or hypotheses. The discussion on the interaction between biofilms and OMV production remains superficial, reducing the impact and depth of the review. Additionally, although potential applications of OMVs are mentioned, these insights are scattered and underdeveloped, making them difficult to follow.
  • To enhance clarity and underscore the novelty of the review, it is recommended that the authors consolidate and expand their discussion of OMV-based applications and strategies to disrupt biofilm formation in a dedicated Discussion section. This would enable a more coherent synthesis of ideas and better emphasize the translational relevance of OMVs, particularly in relation to antimicrobial resistance and therapeutic strategies. Alternatively, each major section of the manuscript i.e. 2. Biofilms in gram-negative ESKAPE bacteria and 3. OMVs in gram-negative ESKAPE bacteria - be revised to incorporate a more critical perspective by systematically outlining: (1) current limitations in understanding, (2) potential avenues for future investigation, and (3) the broader implications of these findings for antimicrobial resistance and therapeutic development. This structured approach would enhance the analytical depth of the review and provide a more valuable resource for researchers in the field.
  • Given the manuscript’s length and density, the authors are encouraged to present some of the detailed information more synthetically using summary tables or schematic figures, that would help readers navigate complex information more efficiently and improve the overall readability and impact of the review.

Specific comments

  • The Title, please clarify the “ESKAPE negative bacteria”
  • Table 1 – please group the antibiotics based on their classes
  • Lines 63-73, while the text introduces important points about the intersection of AMR and COVID-19, it lacks depth and specificity. The text mentions that A. baumannii was frequently detected and that certain pathogens were major causes of VAP, but it does not provide: prevalence rates, comparative data from pre-pandemic periods, statistical significance or confidence intervals, sample size or study design, quantitative data. The findings from Mexico are presented alongside a global systematic review, but the text does not reconcile regional differences or explain why local trends may diverge from global ones. The term “syndemic” is introduced but not unpacked. 

Thank you!

Kind regards!

Author Response

Dear Reviewer 3,

We sincerely appreciate the time you dedicated to reviewing our manuscript and for providing critical observations. While we recognize the value of your comments, some of the points raised would considerably broaden the scope of our work beyond what is feasible within a single article. Since the manuscript is already extensive, we focused our revisions on clarifying potentially misleading sections rather than expanding into areas that would require a monograph-length treatment.

Specifically:

  • The concept of syndemic was used in the introduction based on reliable sources, and the link between COVID-19 infection and increased bacterial resistance was illustrated with an example. We agree that the broader issue of antimicrobial resistance during the COVID-19 pandemic is highly important but also requires a separate, dedicated analysis.
  • Our team is currently preparing a monograph that will address biofilm and OMV formation at the genetic and mechanistic levels, where these broader aspects will be explored in detail. In this context, your insights are valuable, and we would gladly consider potential collaboration in the future.
  • Regarding the article title, the abbreviation ESKAPE is listed in the abbreviations section and explained in the introduction.
  • For clarity, we have revised Table 1 (former Figure 1) and expanded Figure 3 with more detailed explanations.

Once again, we warmly thank you for your thoughtful feedback, which helped us improve the clarity of our manuscript.

With kind regards,

Authors

Round 2

Reviewer 3 Report

Comments and Suggestions for Authors

Dear authors,

Thank you for your revised manuscript and for your thoughtful response to the review comments.  I understand your concern regarding the scope and length of the manuscript, and I acknowledge your intention to address broader mechanistic aspects in a future monograph. That said, for this review to meet the standards of analytical depth expected by the journal, I still recommend that you incorporate a more critical perspective within the current framework. Specifically, without expanding the manuscript significantly, you could:

  • Clarify the experimental context of the cited studies (e.g., in vitro vs. in vivo vs. clinical) to enhance translational relevance.
  • Strengthen the discussion on the interaction between biofilms and OMVs, even briefly, to provide a more integrated view.
  • Consolidate insights on OMV-based applications into a more cohesive paragraph or subsection to improve clarity and impact.

Also, regarding the manuscript length, it is currently quite dense. I recommend reviewing the text for conciseness and considering how some of the detailed information might be more effectively presented through summary tables or schematic figures. These visual elements would not only improve readability but also help highlight key concepts more clearly.

Additionally, I encourage you to create original figures specifically tailored to this manuscript, rather than reusing or adapting visuals from your previously published work. This will ensure that the illustrations are fully aligned with the structure and focus of your current review and meet the standards of originality expected by the journal.

I appreciate the improvements made, particularly the updates to Table 1 and Figure 3, as well as your clarification regarding the use of the ESKAPE acronym. However, several important points still require attention to meet the journal’s standards:

  • Figure 1 remains quite superficial, presenting only the four steps of biofilm development without accompanying explanations or a legend. Enhancing this figure with brief descriptions or a caption would greatly improve its clarity and usefulness.
  • Table 1 has been revised, but contains errors, such as amikacin is incorrectly listed as a beta-lactam antibiotic. Please carefully review and verify the classification of all antibiotics included. There are several major classes of antibiotics, beside beta-lactams, for example ciprofloxacin and levofloxacin are quinolone and so on...
  • Regarding the title, my initial comment specifically requested clarification of the term “ESKAPE-negative bacteria.” ESKAPE pathogens include both Gram-positive and Gram-negative species. The terminology ESKAPE-negative bacteria is misleading and should be revised for accuracy.

While I understand your intention to maintain a focused scope, I believe these adjustments can be made without significantly expanding the manuscript. They will enhance the precision and clarity of your work and ensure it provides maximum value to readers.

I look forward to reviewing your updated version and appreciate your continued efforts to improve the manuscript.

Thank you!

Warm regards,

Author Response

Please find the reply in the document attached. Thank you.

Round 3

Reviewer 3 Report

Comments and Suggestions for Authors

Thank you very much for addressing my comments and suggestions.